# Connectome Mapping: Shape-Memory Network via Interpretation of Contextual Semantic Information

**Kyungsu Lee**
Department of Computer Science & Artificial Intelligence
Jeonbuk National Univesity
`ksl@jbnu.ac.kr`

**Haeyun Lee**
School of Computer Science and Engineering
Korea University of Technology & Education
`haeyun.lee@koreatech.ac.kr`

**Jae Youn Hwang**[*]
Department of Electrical Engineering & Computer Science
Daegu Gyeongbuk Institute of Science and Technology (DGIST)
`jyhwang@dgist.ac.kr`

## Abstract

Contextual semantic information plays a pivotal role in the brain's visual interpretation of the surrounding environment. When processing visual information, electrical signals within synapses facilitate the dynamic activation and deactivation of synaptic connections, guided by the contextual semantic information associated with different objects. In the realm of Artificial Intelligence (AI), neural networks have emerged as powerful tools to emulate complex signaling systems, enabling tasks such as classification and segmentation by understanding visual information. However, conventional neural networks have limitations in simulating the conditional activation and deactivation of synapses, collectively known as the connectome, a comprehensive map of neural connections in the brain. Additionally, the pixel-wise inference mechanism of conventional neural networks failed to account for the explicit utilization of contextual semantic information in the prediction process. To overcome these limitations, we developed a novel neural network, dubbed the Shape Memory Network (SMN), which excels in two key areas: (1) faithfully emulating the intricate mechanism of the brain's connectome, and (2) explicitly incorporating contextual semantic information during the inference process. The SMN memorizes the structure suitable for contextual semantic information and leverages this structure at the inference phase. The structural transformation emulates the conditional activation and deactivation of synaptic connections within the connectome. Rigorous experimentation carried out across a range of semantic segmentation benchmarks demonstrated the outstanding performance of the SMN, highlighting its superiority and effectiveness. Furthermore, our pioneering network on connectome emulation reveals the immense potential of the SMN for next-generation neural networks.

## 1 Introduction

The past few decades have witnessed remarkable progress in deep learning (DL) research, largely driven by the significant advancements in graphics processing units (GPUs). These GPUs, with their exceptional computational powers, have played a pivotal role in accelerating the development of DL methodologies. Fully Connected Networks (FCN) and Visual Geometry Group (VGG) have

been introduced as early baseline networks (Long et al., 2015; Simonyan & Zisserman, 2014). Recently, deep neural networks (DNNs) have been applied to many tasks, such as YoLo for detection tasks (Redmon & Farhadi, 2018) and U-Net for segmentation tasks Ronneberger et al. (2015). More recently, efforts to emulate the neuronal and cognitive intricacies of the human brain have continued to prompt the development of advanced DL models. For instance, Woo et al. (2018) conceptualized the Convolutional Block Attention Module (CBAM), a novel architecture encapsulating the attention mechanism, and Vaswani et al. (2017) introduced the Transformer model processing sequential data. Moreover, an innovative perspective of treating images as sequences by Dosovitskiy et al. (2020) led to the development of Vision Transformer (ViT), which interprets images as sequential data.

Nonetheless, Convolutional Neural Networks (CNNs) and Vision Transformers (ViTs) have limitations when it comes to effectively leveraging contextual semantic information. In contrast, the human brain excels at visually interpreting objects' morphological attributes by actively incorporating contextual semantic information, thereby facilitating a holistic understanding of the surrounding environment (Trobe, 2001; Farah, 2000). For example, the human brain intuitively recognizes that the sky appears above the land, vehicles tend to be found on roads, and the road area is typically more extensive than the space occupied by cars. These contextual semantic cues contribute to our comprehensive understanding of the visual environment (Farah, 2000; Grill-Spector & Malach, 2004). Similarly, artificial intelligence (AI) relies on the utilization of contextual semantic information to accurately identify objects within its environment (Brézillon, 1999; Chalmers et al., 1992). This parallelism with human brain functioning implies that neural networks (NNs), like their biological counterparts, also require the ability to incorporate contextual cognition (Chalmers et al., 1992; Goodfellow et al., 2016; Nebauer, 1998). However, despite recent advancements in NNs that integrate spatial information of objects (Jaderberg et al., 2015; He et al., 2016), there are still constraints in effectively incorporating contextual semantic information. This limitation stems from pixel-level classification approaches that lack a deep comprehension of the morphological attributes of objects (Guo et al., 2022; Zheng et al., 2021). As a result, although NNs demonstrate proficiency in identifying objects based on spatial information, they have faced challenges in comprehending the intrinsic semantics (Waldrop, 2019; Goodfellow et al., 2016; Guo et al., 2022).

In the early era, the pioneering work of Rosenblatt (1958); Minsky & Papert (1988) introduced the concept of multi-layer perceptrons as an attempt to emulate the mechanism of human neurons. Furthermore, the concept of the NNs, interconnected perceptrons designed to mimic the complex mechanisms within the brain, has been introduced (Rumelhart et al., 1985; 1986). Additionally, a range of activation functions have been developed alongside NNs to emulate neurotransmission, including baseline activation functions such as sigmoid and ReLU, as well as advanced functions such as ELU (Clevert et al., 2015), GALU (Hendrycks & Gimpel, 2016), Swish (Ramachandran et al., 2017), SeLU (Klambauer et al., 2017), and ASH (Lee et al., 2022). The activation functions in NNs have been developed to simulate neurotransmission by emulating the mechanisms of membrane and action potentials. However, it is important to note that the human signal transmission system is significantly more complex, incorporating both electrical signals within neurons and chemical signals in synaptic transmission, as depicted in Figure 1. Hence, the previous methods have been limited in simulating complex neural transmission systems. To address this, a novel Spike Neural Network (SNN) has been proposed in recent years (Tavanaei et al., 2019; Lee et al., 2016). However, the lack of an optimal training algorithm for SNNs has been a limiting factor for their practical application in real-world scenarios.

**Contribution**  We present a two-fold objective aimed at (1) devising a novel neuro-mechanical neurotransmission model inspired by the signal transmission processes within the human brain and synapses, implementing an electrochemical methodology, and (2) developing a novel network that explicitly employs contextual semantic information during segmentation. Building upon the insights from the ensemble network (see Section 2), we introduce a novel network architecture that aims to optimize predictive performance. This innovative network dynamically adapts its structure based on the contextual semantic information embedded in the input image during the prediction process. Due to its ability to store and recall the optimal architecture in response to contextual semantic information, we have coined the term Shape-Memory Network (SMN) for this network. Furthermore, to facilitate structural modifications, we designed a novel conditional neuron capable of altering the inter-neuronal connections in response to received control signals that regulate inter-neuron signal transmission and neuronal activation (Fig. 1). In summary, our approach involved the two-fold design of a network that explicitly harnesses and capitalizes on contextual semantic information. First, we

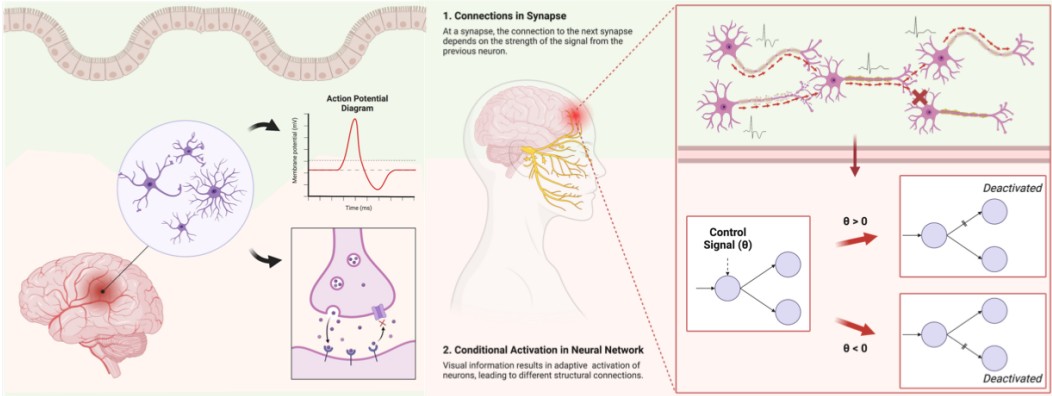

Figure 1: Schematic illustrations depicting the electrochemical neurotransmission process in the human brain, as well as the mechanism by which control neurons emulate neurotransmission. In this mechanism, the architecture of neural connections in control neurons is dynamically modified in response to control signals, resulting in a variable network architecture.

proposed a conditional neuron with a novel signal transmission system facilitating structural variants upon the control signal. Second, this conditional neuron was then organically integrated into the SMN.

The main contributions of this paper are summarized below:

- We proposed a novel network (i.e., SMN) that explicitly leverages the contextual semantic information for segmentation tasks by adjusting its architecture in a test-time fine-tuning manner and providing an appropriate structure for a particular domain.

- To realize the explicit utilization of contextual semantic information, we developed a novel algorithm that reconstructs an entropy-map by integrating CAMs (Zhou et al., 2016).

- In SMN, we introduced a novel mechanism, the conditional neuron, incorporating a control signal and enabling contextually adaptive activation of inputs, drawing inspiration from synaptic mechanisms while maintaining its distinct computational characteristics.

- Through rigorous mathematical justification, we provided a solid theoretical foundation for the concept of conditional neurons within the SMN. Furthermore, to validate the practical performance of the SMN, we extensively tested it on multiple segmentation benchmark datasets, demonstrating its superior segmentation capabilities compared with other methods.

## 2 PROBLEM STATEMENT

**Semantic Domain Gap**    Many studies have highlighted the effect of domain gaps hampering the predictive efficacy of DL networks (Shu, 2015; Wei et al., 2018). The domain gap significantly emerges due to the diversity in sensors, environmental conditions during data acquisition, or variations in pre-processing methodologies, particularly in the field of computer vision (Regmi & Shah, 2019; Nam et al., 2021). The datasets, each with characteristic attributes, are typically considered distinct domain groups, yet Pan et al. (2020) introduced the existence of intra-group domain gaps within one single domain as well. In this work, we aim to focus on contextual semantic information as a primary one of the multiple factors contributing to domain gaps. Contextual semantic information incorporates semantic attributes of objects such as their size (proportional area occupied in the image, denoted as density in this paper), spatial location, and morphological form.

Suppose three subsets of $U_1$, $U_2$, and $U_3 \subset U$ in the multi-dimensional space of contextual semantic information ($U \subseteq \mathbb{R}^{H \times W \times C}$). Suppose $U_1$ and $U_2$ are similar, and $U_1$ and $U_3$ are different in terms of contextual semantic information, such that $|U_1 - U_2| < |U_1 - U_3|$ where $|A - B|$ indicate the average distance between all samples in the sets of $A$ and $B$. Additionally, let $\mathcal{L}(\theta^M; U)$ be a loss function that leverages the samples in $U$, using the parameters ($\theta^M$) of a DL model ($M$), leading to

**Proposition I.** $|U_1 - U_2| < |U_1 - U_3| \implies \mathcal{L}(\Theta^M; U_2) < \mathcal{L}(\Theta^M; U_3)$ where $\Theta^M = \operatorname{argmin}_{\theta^M} \mathcal{L}(\theta^M; U_1)$.

The **Proposition I** implies that the DL model optimized to a specific domain provides imprecise predictions on the different domains. Thus, the problem statement that aims to find the DL network providing precise predictions disregarding domain gaps is formulated as below:

$$\Theta^M = \underset{\theta^M}{argmin}\mathcal{L}(\theta^M; U_1) \text{ and } |U_1 - U_2| < |U_1 - U_3| \implies \mathcal{L}(\Theta^M; U_2) \sim 0 \text{ and } \mathcal{L}(\Theta^M; U_3) \sim 0 \quad (1)$$

**Ensemble Model**  The trivial solution to bridge the domain gaps and achieve precise predictive accuracy is to employ an ensemble DL model for the prediction. Suppose different sets of $U_i$ where $(i = 1, 2, ..., N)$ indicating different $N$ numbers of domains and DL models ($M_i$), which is optimized to $U_i$, respectively. Then the ensemble model can provide precise prediction as below:

**Proposition II.** To precisely predict sample ($u \in \bigcup_i U_i$), the ensemble model ($M$) can be derived using models optimized for each domain as $M(u) = M_i(u)$ (if $u \in U_i$).

Therefore, by employing an appropriate model suitable to each domain, the ensemble model can provide precise predictions and thus realize Equation 1. However, note that the ensemble model exhibits limitations, including heavy memory requirements and the inability to provide accurate predictions for domains that have not been trained.

**Optimal Solution**  Many domain adaptation (DA) methodologies have been extensively studied to mitigate domain gaps, encompassing various approaches such as transfer learning (Patricia & Caputo, 2014; Kouw & Loog, 2018), generative DA (Bousmalis et al., 2017; Hu et al., 2018), and unsupervised and self-supervised DA (Pan et al., 2020; Xu et al., 2019; Liu et al., 2021; Bartler et al., 2022). Among them, test-time adaptation (TTA) has emerged as a prominent approach, similar to the ensemble model for addressing Equation 1, where the TTA models yield improved prediction accuracy by retraining the network with optimized parameters during the inference phase (Liu et al., 2021; Bartler et al., 2022). In this work, we integrate the ensemble model with the TTA method to tackle Equation 1 (Liu et al., 2021; Bartler et al., 2022). Consequently, we propose a DL model that fine-tunes its parameters and dynamically adjusts the optimal architecture.

## 3 METHOD

### 3.1 SHAPE-MEMORY NETWORK

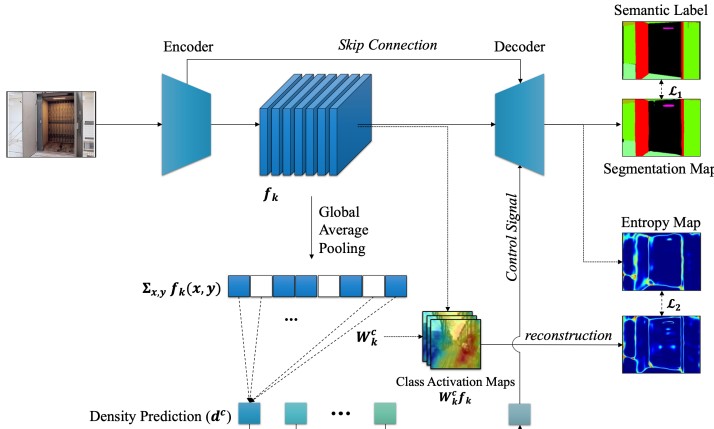

Figure 2: Illustration of the architecture of the SMN, consisting of several components, including segmentation, density regression, entropy map reconstruction, and signal control.

**Design Principle**  The fundamental architecture of our Shape-Memory Network (SMN) is designed to process and utilize contextual semantic information in visual data effectively. For instance, consider a semantic segmentation task on urban scene datasets. The input images typically contain multiple

object classes with consistent spatial and contextual information: transportation infrastructure (roads, sidewalks) occupies the lower regions, architectural structures appear with specific scale constraints, and environmental elements (sky, vegetation) maintain consistent spatial positions.

To implement this, the SMN captures the contextual patterns via two primary computational components. First, the component implements density mapping, quantifying the proportional distribution of object classes within the input space. Particularly, in urban scene analysis, road surfaces typically constitute 30-40% of the pixel space, while vehicular objects occupy 5-10%. The density distributions are represented as statistical priors, leading to the network for validating segmentation predictions against expected contextual patterns. Significant deviations from the learned distributions (e.g., vehicles occupying 80% of the pixel space) are automatically flagged as anomalous configurations. The second component facilitates entropy mapping, quantifying information complexity in the spatial regions. The entropy mapping mechanism is particularly important for analyzing regions with high-class intersection probability, such as object boundaries or regions of class ambiguity. Computationally, regions exhibiting higher entropy values indicate areas requiring more sophisticated feature extraction and analysis than regions with uniform class distribution.

**Architectural Design**     Regarding the design principles, we formalize our SMN structure with several key mathematical components. Particularly, the SMN employs conditional neurons to transform its structure during test-time adaptation (TTA) dynamically. Furthermore, we implement a self-supervised learning-based re-optimization method, utilizing the entropy-map as a medium for loss minimization and explicit integration of contextual semantic information. While spatial information is effectively conveyed through skip connections, we focus on optimizing the network's contextual understanding by introducing density measurements that quantify the proportional distribution of object classes. Therefore, we focus on optimizing the network's insight into contextual semantic information of input images by introducing density, representing the proportion of the occupied area in the image.

**Definition I.** Let $\Omega_c(h, w; I)$ be a category ($c$) recognition function at pixel $I\|_{h,w}$ in input ($I$), such that $\Omega_c(h, w; I)$ is 1 *iff* $\underset{x}{\mathrm{argmax}}\, I\|_{h,w} = c$, otherwise 0.

**Definition II.** Let $d_l^c : \mathbb{R}^{H \times W \times C} \to \mathbb{R}$ be the density function of the target object ($c$) in semantic label ($\hat{y} \in \mathcal{Y} \subset \mathbb{R}^{H \times W \times 3}$), such that $d_l^c(\hat{y}) = \frac{1}{HW} \sum_h^H \sum_w^W \Omega_c(h, w; I)$ with the image of height ($H$), width ($W$), and the number of categories ($C$).

**Lemma I.** $\sum_c^C d_l^c(\hat{y}) = 1$ since $\sum_c^C \sum_h^H \sum_w^W \Omega_c(h, w; I) = HW$.

The density-regression pipeline facilitates two functions: (1) it enables the generation of Class Activation Maps (CAM) and entropy-maps for TTA optimization, and (2) it manages control signals for structural transformation based on input characteristics. By leveraging our novel approach, the CAM not only captures the visual attributes of target objects but can also be transformed into an entropy-map. This allows us to optimize the SMN, by minimizing the similarity loss between the entropy-map reconstructed using CAM and the entropy-map generated in the segmentation pipeline.

**Network Architecture**     Fig. 2 illustrates the detailed architecture of the SMN based on multi-task and self-supervised learning for TTA. The SMN incorporates the segmentation pipeline and the density-regression pipeline. Here, the main task of the SMN is the semantic segmentation task to localize objects into segmentation maps, and the pretext task is to predict the density of the target object in a multi-labeled manner. Note that the cognition of the contextual semantic information could be explicitly realized by incorporating the spatial information conveyed from the skip connections and the mathematical morphology achieved by the recognition of density prediction. Subsequently, the cognition of contextual semantic information leads to the structural transformation of the SMN via the entropy-map-based optimization in a self-supervised and TTA manner.

Let $\mathcal{X} \subset \mathbb{R}^{H \times W \times 3}$ and $\mathcal{Y} \subset \mathbb{R}^{H \times W \times C}$ be the sets of input RGB images and corresponding segmentation labels, where $H$ and $W$ are the height and width of an input image, and $C$ is the number of categories of inputs, and $E : \mathbb{R}^{H \times W \times 3} \to \mathbb{R}^{H' \times W' \times k}$ and $D : \mathbb{R}^{H' \times W' \times k} \to \mathbb{R}^{H \times W \times C}$ the encoder and decoder of SMN, respectively, where $\mathbb{R}^{H' \times W' \times k}$ is the encoded feature space for $f_k$ in Fig. 2, such that $M(x) := (D \circ E)(x)$ and $f_k := E(x)$ where $x \in \mathcal{X}$ and $k$ is the number of channels. Subsequently, assuming the predicted segmentation map ($M(x)$) represents the probabilities obtained from the softmax output, the following constraint is imposed on the subsequent operations.

**Lemma II.** $0 \leq M(x)\|_{h,w,c} \leq 1$, and thus $\sum_c M(x)\|_{h,w,c} = 1$.

Therefore, in the predicted segmentation map and density-regression pipeline, the density of the target object is defined as below:

**Definition III.** Let $d_s^c : \mathbb{R}^{H \times W \times 3} \to \mathbb{R}$ be the density function of the target object ($c$) on $x \in \mathcal{X}$ using the SMN, such that $d_s^c(x) = \frac{1}{HW} \sum_h^H \sum_w^W \Omega(h, w; M(x))$ and $\sum_c d_s^c(x) = 1$ by **Lemma II**.

**Definition IV.** Let $d^c : \mathbb{R}^{H \times W \times 3} \to \mathbb{R}$ be the density function of the target object ($c$) predicted density on $x \in \mathcal{X}$ using the SMN, such that $d^c(x) = \sum_k W_k^c \sum_{h,w} f_k(h, w)$.

**Optimization and Inference of SMN** The SMN is trained via three loss functions. Like in the general segmentation task, the predicted segmentation map ($\hat{y} = (D \circ E)(x)$ where $x \in \mathcal{X}$) by the SMN is optimized to the segmentation label ($y \in \mathcal{Y}$) via the cross-entropy loss function ($\mathcal{L}_{\text{CE}}$), such that $\mathcal{L}_1(\theta^M; (\mathcal{X}, \mathcal{Y})) := \sum_x^{\mathcal{X}} \mathcal{L}_{\text{CE}}(y, \hat{y})$. Additionally, the predicted entropy-map in the segmentation pipeline is optimized to the entropy-map reconstructed from the CAM, such that $\mathcal{L}_2(\theta^M; \mathcal{X}) := \sum_x^{\mathcal{X}} \mathcal{L}_{\text{ssim}}(\mathbb{E}(x), EM(\sum_k W_k^c f_k))$, where $\mathbb{E}(x) = -\sum_c M(x)\|_{h,w,c} \log M(x)\|_{h,w,c}$, $\mathcal{L}_{\text{ssim}}$ is the structural similarity loss (Lu, 2019), and $EM$ is the entropy-map-reconstructing algorithm from the CAM (See Sec. 3.2). The $\mathcal{L}_2$ implies explicitly utilizing contextual semantic information during segmentation. Furthermore, the predicted density ($d^c$) by the SMN is optimized to the calculated density by the predicted segmentation map based on **Definition II** and **Definition IV**, such that $\mathcal{L}_3(\theta^M; (\mathcal{X}, \mathcal{Y})) := \sum_x^{\mathcal{X}} |d^c(x) - d_l^c(y)|_2$. Therefore, the following constraint is imposed on the subsequent operations.

**Proposition III.** $d^c$ in the density-regression pipeline of the optimized SMN satisfies the property in **Lemma I**, such that $0 \leq \sum_k W_k^c \sum_{h,w} f_k(h, w) \leq 1$ and $\sum_c \sum_k W_k^c \sum_{h,w} f_k(h, w) = 1$ by **Lemma II**.

**Proposition IV.** The optimization of $d_c$ facilitates the incorporation of contextual semantic information into the parameters of the SMN, enhancing its ability to comprehend and utilize such information effectively.

During the inference phase, the SMN is re-optimized via a self-supervised manner only using $\mathcal{L}_2$. Particularly, $\mathcal{L}_2(\theta'^M; \mathcal{X})$ is applied, where $\theta'^M$ represents the parameters for the condition signals, thus indicating the manifestation of the structural transformation of the SMN at the inference phase. The TTA employing $\mathcal{L}_2$ yields two primary outcomes: (1) TTA enables expedited re-optimization and minimizes memory requirements by utilizing a reduced number of training parameters, and (2) TTA enhances prediction accuracy by providing an optimal architecture aligned with the contextual semantic information of input $x$.

## 3.2 ENTROPY-MAP RECONSTRUCTION VIA CLASS ACTIVATION MAP

In the previous work, Zhou et al. (2016) introduced the CAM ($\sum_k W_k^c f_k$), directly indicating the importance of the activation at spatial grid $(h, w)$. Additionally, $d^c(x)$ indicates the density of target objects in $x$ (e.g., $d^c(x) = 0$ implies that the absence of $c$ in $x$ and $d^c(x) = 1$ implies that the $x$ is filled with $c$). Since we designed and trained $0 \leq d^c(x) = \sum_k W_k^c \sum_{h,w} f_k(h, w) = \sum_{h,w} \sum_k W_k^c f_k(h, w) \leq 1$, Based on this, we expect the important area in density regression to be the contextual semantic information in terms of mathematical morphology, and thus $\sum_k W_k^c f_k$ highlights the regions of the target object ($c$), and $\sum_k W_k^c f_k(h, w)$ implies the expected density at the spatial grid of $(h, w)$.

**Proposition V.** The CAM ($\sum_k W_k^c f_k$) generated by the SMN is figured to highlight the regions of $c$.

Moreover, by stacking the CAMs for each category ($c$) as $[\sum_k W_k^c f_k(h, w)] \in \mathbb{R}^C$, the concatenated feature-map incorporates the important ratio for density prediction of each category $c$, and thus the factors could be normalized by using the softmax function to calculate stochastic variables $\bar{d}^c(x)\|_{h,w}$ for the density of target $c$, and thus $\sum_c \bar{d}^c(x)\|_{h,w} = 1$. Therefore, the entropy-map is calculated by leveraging the probability as $-\sum_c \left( \bar{d}^c(x)\|_{h,w} \log \bar{d}^c(x)\|_{h,w} \right)$, and the **Theorem** for the definition of $EM(x)$ is formulated.

**Theorem I.** $EM(x)\| = -\sum_c \left( \bar{d}^c(x)\|_{h,w} \log \bar{d}^c(x)\|_{h,w} \right)$ where $\bar{d}^c(x)\|_{h,w} = \frac{e^{\sum_k W_k^c f_k(h,w)}}{\sum_c e^{\sum_k W_k^c f_k(h,w)}}$

### 3.3 CONTROL NEURON

The control neuron functions as a fundamental element within the adaptive architecture of the SMN. It processes information through three interconnected pipelines that collectively define its operation: (1) standard neural inputs from linked neurons analogous to those in traditional neural networks, (2) a control signal based on predicted density distributions, and (3) a self-activation mechanism gating signals. The three pipelines enable the network to dynamically adapt its structure in response to varying input characteristics, resembling how biological neural systems adjust connectivity patterns. During the processing of an input image, control neurons selectively engage or disengage connections based on contextual information, thereby achieving an optimal configuration for the specific input.

As **Proposition IV**, optimizing $d^c$ promotes the interpretation of contextual semantic information. Consequently, employing $d^c(x)$ as a control signal enables the structural transformation of the SMN by aligning its structure with the contextual semantic information. For instance, to achieve **Proposition II**, suppose three inputs of $x_1$, $x_2$, and $x_3 \in \mathcal{X}$, and let $d^c(x_1)$ and $d^c(x_2)$ be in a similar density distribution $(d^c(x))$, whereas $d^c(x_3)$ be in a different distribution, such that $D_B(d^c(x_1), d^c(x_2)) < D_B(d^c(x_1), d^c(x_3))$, where $D_B(P, Q)$ is Bhattacharyya distance. Additionally, suppose two structurally distinct networks of $M_1$, which yields optimal performance for $x_1$ and $x_2$, and $M_2$ for $x_3$. Then, the SMN predicts $x_1$, $x_2$, and $x_3$ as below:

$$M(x) = \begin{cases} M_1(x) \text{ (if } x = x_1 \text{ or } x = x_2) \\ M_2(x) \text{ (if } x = x_3) \end{cases} \tag{2}$$

Note that, the control signal affects the selection of optimal $M_i$ in Equation 2. To design a control signal that interprets the contextual semantic information, we employed the predicted densities $(d^c)$ as inputs based on **Proposition IV**, and formulated the control signal as below.

**Definition V.** The control signal of SMN is obtained by a linear combination of predicted densities $(d^c)$.

To implement **Definition V**, the densely connected parameters were employed in the SMN (Fig. 2). Suppose $[d^c(x)]_{c=1,2,...,C} \in \mathbb{R}^{1 \times C}$ and the parameters of $[[v_c]_{c=1,...,C}]_{n=1,...N_{cn}} \in \mathbb{R}^{C \times N_{cn}}$, where $N_{cn}$ is the number of control neurons. Subsequently, we defined the linear combinations of $d^c(x)$ and $v_n$ as $\sum_c^C v_{n,c} d_c(x) \in \mathbb{R}$ as the control signal of $n^{th}$ control neuron. Note that, the parameter of $v$ is trainable, and thus the architecture of the SMN is aligned alongside the contextual semantic information of input $(x)$ upon the predicted density in the train and inference phases.

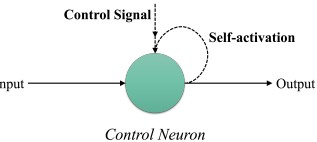

Figure 3: control neuron.

Furthermore, to effectuate the structural transformation in response to the control signal, each neuron in the SMN is designed to receive three distinct input signals of inter-neuronal inputs, self-activation, and a control signal that incorporates contextual semantic information resampled from the density distribution (**Definition V**), as depicted in Fig. 3. Similar to the human brain, wherein the inter-neuronal transmission is facilitated via chemical signal transmission in synapses, inputs from other neurons in the SMN are designed to emulate the chemical neurotransmission. Additionally, the control signal and self-activation are conceived to emulate the electrical transmission mechanism. Consequently, the control signal is intended to simulate the threshold for membrane potential, and self-activation is associated with the sustained stimulus to neurons. Furthermore, the output of the control neuron emulates the action potential.

**Definition VI.** The output of a control neuron is $\mathcal{A}(\sum \text{Input}_i) * ((\text{Control} > t_n) \mid (\text{self-activation}))$ where $\mathcal{A}$ is an activation function, $t_n$ is a neural threshold of $n^{th}$ control neuron of SMN, $*$ is a arithmetic multiplication operator, and | is bit-wise or operator.

To design an output responsive to a control signal and neural threshold, the terms of control signal and self-activation are implemented into the $\mathcal{A}(\sum \text{Input}_i)$, which is the same output of the conventional neuron. The output signal is activated when the amplitude of the control signal surpasses an intrinsic threshold or when self-activation is true. The activation of the control signal indicates that the neuron has been subjected to a stimulus exceeding the threshold, and self-activation is introduced to prevent the loss of informative activation caused by the sparsity problem. The criteria for informative activation derive from the premise that among interconnected neurons (feature maps),

a specific neuron hold more information than others. Let $s_i^{\text{in}}$, $s^{\text{out}}$, and $s^{\text{ct}}$ be input signals, output signal, and control signal, and $\lambda_n$ and $\mathcal{A}$ be the intrinsic threshold of neuron and the adaptive activation function filtering informative activation (*Appendix*), then **Definition VI** is formulated as: $s^{\text{out}} = \left( H(s^{\text{ct}} - \lambda_n) + H(\mathcal{A}(\sum_i s_i^{\text{in}})) \right) \sum_i s_i^{\text{in}}$, where $H(x)$ is a Heaviside step function, where $H(x) = \max(x, 0)$. However, since the $\lambda_n$ is not arithmetically connected to input signals, the $\lambda_n$ is not trainable, but the intrinsic threshold of $\lambda_n$ should be trainable. To realize, we approximated the $H(x)$ into arithmetic form, and thus the output of the control neuron is formulated using a sigmoid function ($\sigma(x)$) and a large value of $\alpha$ as

**Theorem II.** The output signal of the control neuron with input ($s_i^{\text{in}}$) and control ($s^{\text{ct}}$) signals is formulated as $\left( \sigma(-2\alpha_1(s^{\text{ct}} - \lambda_n)) + \sigma(-2\alpha_2 \mathcal{A}(\sum_i s_i^{\text{in}})) \right) \sum_i s_i^{\text{in}}$.

Since the $\lambda_n$ is arithmetically connected to the inputs, the $\lambda$ is trainable during the training phase. Therefore, the control neurons retain the intrinsic threshold to store the optimal architecture of the SMN, and the SMN adjusts its own architecture by fine-tuning $v$ in run-time to align the appropriate architecture to contextual semantic information.

## 4 MAIN RESULTS

To evaluate the performance of SMN, various benchmark datasets, such as aerial imagery datasets of Inria (Maggiori et al., 2017b) and LoveDA (Wang et al., 2021) and scene understanding benchmarks of ADE20K (Zhou et al., 2017), Youtube-VOS (Xu et al., 2018), and BDD100K (Yu et al., 2020) were utilized. To demonstrate the general feasibility of the SMN for semantic segmentation, the datasets for scene understanding were employed. In addition, since the density of the objects should be a crucial feature for aerial imagery, aerial datasets are employed to illustrate the strength of the SMN effectively. Furthermore, the GTA5 dataset (Richter et al., 2016) was utilized to evaluate the scalability of the SMN to the synthesis parsing. The more detailed descriptions of the datasets are illustrated in the *Appendix*.

### 4.1 COMPARISON ANALYSIS

To compare the segmentation performance of the SMN, we employed compatible DL models, such as the baseline models (Ronneberger et al., 2015; Xie et al., 2021), multi-Path models (Zhuang, 2018; Bai & Zhou, 2020), state-of-the-art for segmentation models (Seg-SotA) (Wang et al., 2022b;c), and video object segmentation models (VOS) Cheng & Schwing (2022); Yang et al. (2022). To evaluate the efficacy and superior performance of the SMN in the segmentation task, the baseline models, the SotA models, and the VOS models were employed. Additionally, the performance of SMN was compared using the multi-path models, which stand a similar role to the ensemble model. The detailed descriptions for the datasets and experimental setups are illustrated in the *Appendix*.

Table 1: Quantitative comparison analysis of SMN to other compatible deep learning models in terms of intersection over union (IoU). The best performance values are highlighted in **bold**, and the second-best values are underlined.

|  | Baseline Model | | Multi-Path | | Seg SotA | | VOS | | Ours | |
|---|---|---|---|---|---|---|---|---|---|---|
|  | U-Net | SegFormer | LADDERNet | MPDNet | InternImage | BEiT-3 | Xmem | AOST | Ours - SA | Ours |
| Inria | 62.96% | 67.97% | 64.77% | 64.51% | 68.60% | 66.69% | 64.85% | 69.30% | 68.60% | **72.72%** |
| LoveDA | 47.71% | 51.33% | 49.66% | 48.25% | 49.81% | 49.63% | 51.29% | 50.57% | 50.40% | **54.28%** |
| ADE20K | 42.61% | 46.72% | 52.66% | 44.30% | 51.04% | 51.67% | 44.88% | 52.06% | 48.84% | **55.76%** |
| Youtube-VOS | 77.12% | 81.07% | 86.04% | 83.57% | 85.13% | 86.67% | 83.36% | 87.09% | 85.66% | **88.66%** |
| BDD100K | 36.69% | 42.59% | 41.13% | 40.03% | 47.25% | 39.85% | 42.69% | 43.09% | 43.81% | **48.83%** |
| GTA5 | 65.84% | 65.99% | 68.64% | 70.69% | 70.94% | 70.90% | 68.00% | 71.06% | 69.07% | **76.58%** |

**Quantitative Analysis** Table 1 illustrates the segmentation performance of the SMN compared to other deep learning models. Here, Ours-SA indicates the SMN without a self-activation path in control neurons. Table 1 involves two novel findings of (1) the SMN exhibits a superior segmentation performance than other compatible DL networks, and (2) the self-activation incorporates a significant role in the SMN. The SMN provides the 6.36% improved IoU values on average, and also exhibits a powerful improvement with a 13.15% improvement in maximum. Additionally, depending on the

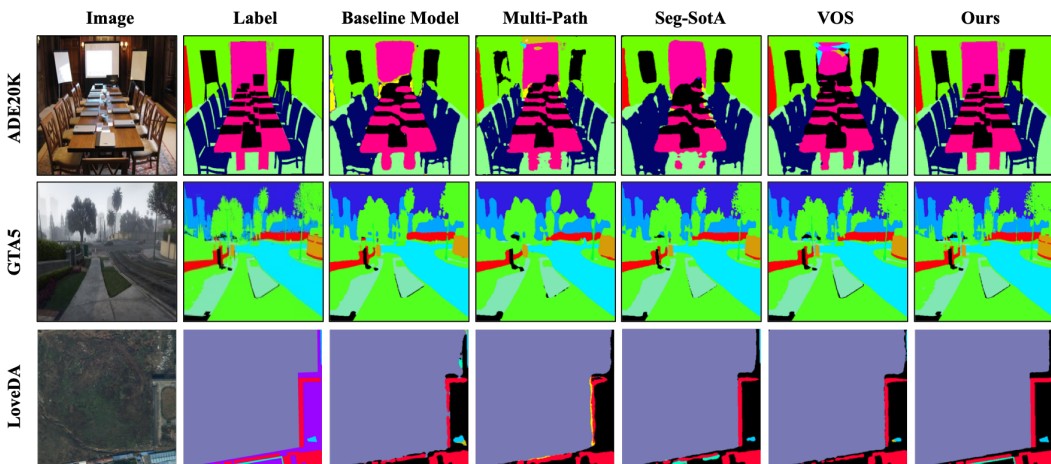

Figure 4: Predicted segmentation maps by SMN (ours) and other competitive DL models.

existence of self-activation, a performance difference of up to 7.51% is exhibited, indicating that self-activation drives a significant contribution to the feature extraction in SMN, implicating that self-activation plays a compulsory role in improving the sparsity problem.

**Quantitative Analysis** Fig. 4 exhibits the segmentation results of SMN and other compatible deep learning models using ADE20K, GTA5, and LoveDA datasets. Upon inspecting the semantic segmentation map predicted by SMN and other DL models, it is evident that the segmentation facilitated by the SMN is remarkably predicted in precise, contrasting with other comparative networks. The quantitative analysis demonstrated the superior performance of the SMN in the segmentation task. Particularly, it is noteworthy that the SMN provides precise segmentation maps when segmenting small objects and objects of various sizes, distinguishing the SMN from other networks by precisely segmenting the boundaries of the objects. Therefore, the segmentation performance of the SMN, when compared to other competitive DL models, was corroborated to stand out in aerial images wherein the density of the objects significantly impacts the segmentation performance. Moreover, the exceptional adaptability of the SMN, highlighted by its ability to adjust its network structure based on contextual semantic information dynamically, further underscores its superiority in delivering precise segmentation, even under diverse and challenging conditions.

## 4.2 EXPLICIT UTILIZATION OF CONTEXTUAL SEMANTIC INFORMATION

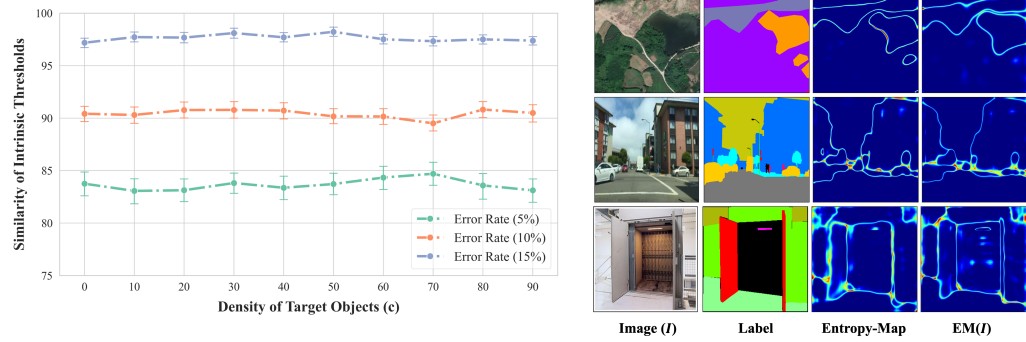

Figure 5: (Left) Similarity of the intrinsic threshold of control neurons containing the similar density of the target objects and (Right) samples of entropy-map and reconstructed entropy-map.

Fig. 5 offers an explicit depiction of contextual semantic information during segmentation by the SMN. Fig. 5(Left) manifests the degree of similarity between the intrinsic threshold values of the control neurons. These neurons are engaged when predicting two disparate images, exhibiting analogous object densities. The premise for determining similarity rests on acknowledging a match when the concurrence level of a specific control neuron, operating in two distinct structures for the prediction of two different images, falls beneath a defined error rate. Therefore, a reduced error rate in similarity computation signifies imposing more stringent conditions. Nonetheless, Fig. 5(Left) highlights a strong correlation between the intrinsic threshold values based on density, and thereby indicating that the structural transformations within the SMN are relevant and fluctuate in response to the contextual semantic information. Fig. 5(Right) illustrates the similarity between the entropy-map generated via the EM algorithm in the optimized SMN, and the entropy-map conveyed within the segmentation pipeline. The experimental results emphasize using contextual semantic information by the SMN during the segmentation tasks.

## 5 DISCUSSION

**Hyper-parameter Tuning**   Note that the experiment did not aim to find the best-performing model with the fully searched parameters, but aimed to demonstrate the feasibility of the proposed deep neural networks and to show superior performance compared to other state-of-the-art (SotA) models. To search for the best parameters for the highest performance remained as future works. Furthermore, among the trainable variables, $z_k$, $alpha_1$, and $\alpha_2$ are not significantly treated in this paper since those values cannot significantly affect the segmentation performance of the SMN. For instance, the values of $\alpha$ are optimized to nearly 10.0, which significantly approximates the sigmoid function to the Heaviside step function. Additionally, the $z_k$ is a hyperparameter of the ASH activation function, such that its values are significantly different by the convolution operations and the locations. Therefore, it is natural that the self-optimization of $z_k$ could lead to the optimal performance of the SMN.

**Computational Complexity**   To implement the SMN for real-world applications, we address the computational complexity of the TTA mechanism. The current implementation requires optimization of matrix $M \in \mathbb{R}^{C \times N}$ during inference, with time complexity $O(T \cdot C \cdot N)$ and space complexity $O(C \cdot N)$, where $T$ represents optimization steps (typically $T \leq 5$), $C$ denotes categories, and $N$ indicates control neurons. While our current implementation achieves 32.8 FPS with 47.5M parameters and 549.8G FLOPs, we propose several optimization strategies to enhance efficiency. These include early stopping criteria ($\mathcal{L}t + 1 - \mathcal{L}t < \epsilon$), parameter pruning ($Mpruned = M \odot (|M| > \tau)$), and quantization ($Mquant = \text{round}(M \cdot 2^b)/2^b$). Preliminary experiments suggest these optimizations could reduce computational overhead by 30-40% while maintaining performance within 1-2% of current results. Future work will focus on developing lightweight TTA variants and memory-efficient implementations to further improve real-time performance.

## 6 CONCLUSION

In this paper, we proposed a novel deep learning network that emulates the brain connectome, which incorporates intricate neural connections, and explicitly leverages contextual semantic information during segmentation. To this end, during the training phase, the network is designed to memorize the optimal structural configurations for contextual semantic information, and to transform into the optimal structure suitable for the input's contextual semantic information during the prediction phase, leading to accurate predictions. To depict the explicit utilization of contextual semantic information for segmentation, we designed a novel optimization method based on the class activation map and entropy-map as illustrated in **Theorem I**. Moreover, to implement the structural transformation of the network, we proposed a novel neuronal system called a control neuron, illustrated in **Theorem II**. To evaluate the performance of the proposed network, we employed several semantic segmentation benchmark datasets, and the experimental results demonstrated the superior predictive performance of our method in the segmentation task. Furthermore, our research is foundational for the next-generative networks capable of emulating the human signal transmission system by incorporating test-time adaptation methods and structural transformation, demonstrating the potential of the SMN for scalability in various applications.

## 7 ACKNOWLEDGEMENT

This work was partially supported by the 2024 innovation base artificial intelligence data convergence project project with the funding of the 2024 government (Ministry of Science and ICT) (S2201-24-1002), and by the Korea Research Institute for Defense Technology Planning and Advancement (KRIT) through the Korea Government (MND) under Grant (R230206). This work was partially supported by the National Research Foundation of Korea (NRF) grant funded by the Korea government Ministry of Science and ICT (MSIT) (No. RS-2024-00358888).

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

## A SHAPE-MEMORY NETWORK

**Mechanism**    To illustrate the mechanisms of the Shape-Memory Network (SMN), we first define the mathematical notations and expressions. Let $\mathbb{E}$ and $\mathbb{D}$ be the encoder and decoder of the SMN, respectively. Then, the encoded feature-map $(f_k)$ would be represented as $\mathbb{E}(I_i) = [f_k]_{k=1,2,...,\mathcal{C}}$, where $I_i \in I \subset \mathbb{R}^{H \times W \times 3}$ is the input image with its height $(H)$ and width $(W)$, and $\mathcal{C}$ is the number of feature-map, and the final output, the segmentation map $(Y_i \in Y \subset \mathbb{R}^{H \times W \times C})$, is represented as $Y = \mathbb{D}([f_k]) = (\mathbb{D} \circ \mathbb{E})(I)$, where $C$ is the number of categories in the datasets. In the pipeline for the class activation map (CAM), the encoded feature-maps are average-pooled, such that $\sum_{x,y} f_k(x, y)$ represents an individual feature. Here, suppose $W_k^c$ for the weight for the density-regression for a category $(c)$, and then density prediction$(d^c : \mathbb{R}^{H \times W \times 3} \to \mathbb{R})$ is calculated as $d^c(I_i) = \sum_k W_k^c \sum_{x,y} f_k(x, y) = \sum_k W_k^c \sum_{x,y} \mathbb{E}(I_i)$, such that $[d^c(I_i)]_{c=1,2,...,C} \in \mathbb{R}^C$. Furthermore, the predicted densities for each category are mapped to the control signal $(s^{\text{ct}} \in \mathbb{R}^{\mathbb{N}})$ with the number of individual pixels of feature-maps in $\mathbb{D}$ $(\mathbb{N})$ via dense layers with the trainable matrix $(\mathcal{M} \in \mathbb{R}^{C \times \mathbb{N}})$ such that $s^{\text{ct}} = [d^c(I_i)] \cdot \mathcal{M}$, where $\cdot$ is the matrix multiplication. Subsequently, the $s^{\text{ct}}$ is imported into the $\mathbb{D}$, and thus the final prediction of the SMN is implemented in detail as below:

$$
\begin{aligned}
Y_i &= \mathbb{D}(\mathbb{E}(I_i); s^{\text{ct}}) \\
&= \mathbb{D}\Big(\mathbb{E}(I_i); [\sum_k^{\mathcal{C}} W_k^c \sum_{x,y} \mathbb{E}(I_i)] \cdot \mathcal{M}\Big)
\end{aligned}
\tag{3}
$$

Note that all elements of the segmentation map $(Y_i)$ are the softmax output, such that $0 \leq Y_i\|_{h,w,c} \leq 1$. Therefore, we can define the trainable parameters of the SMN as (1) parameters of encoder and decoder, such that $\theta_{\mathbb{E}}$ and $\theta_{\mathbb{D}}$; (2) parameters $(W_k^c)$ for the CAM pipeline as a dense layer; (3) matrix $(\mathcal{M})$ to map the predicted density to a control signal. Note that only the matrix $\mathcal{M}$ is optimized in the inference phase to change the structure of the SMN. To summarize, the key outputs by the SMN are listed as below:

$$
\begin{aligned}
\text{Segmentation Map: } Y_i &= \mathbb{D}\Big(\mathbb{E}(I_i); [\sum_k^{\mathcal{C}} W_k^c \sum_{x,y} \mathbb{E}(I_i)] \cdot \mathcal{M}\Big) \\
\text{Density-regression: } d^c(I^i) &= \sum_k W_k^c \sum_{x,y} \mathbb{E}(I_i) \\
\text{Class Activation Map: } \mathbf{C}^c(I^i) &= \sum_k W_k^c f_k = \sum_k W_k^c \mathbb{E}(I^i) \\
\text{Entropy-map: } \mathcal{E}(I^i) &= -Y_i \log Y_i
\end{aligned}
\tag{4}
$$

In addition, as illustrated in Sec.3.2 in the manuscript, remember that the entropy-map function $(\text{EM}(I^i))$ reconstructs the entropy-map from the CAMs ($\mathbf{C}^c$ for $c = 1, 2, ..., C$) as below:

$$
\text{EM}(I^i)\|_{h,w} = -\sum_c^C \bar{d}^c(I^i)\|_{h,w} \log \bar{d}^c(I^i)\|_{h,w} \text{ where } \bar{d}^c(I^i)\|_{h,w} = \frac{e^{[\sum_k W_k^c \mathbb{E}(I^i)]\|_{h,w}}}{\sum_c^C e^{[\sum_k W_k^c \mathbb{E}(I^i)]\|_{h,w}}}
\tag{5}
$$

The training process of the SMN is illustrated in Sec 3.1 of the manuscript. The predictive procedure of the SMN in the inference phase is listed as the following:

1. The SMN generates the pseudo-labels for the segmentation map and density-regressions.

2. The entropy-maps are generated via the pipeline of the segmentation $(\mathcal{E}(I^i))$ and the reconstruction algorithm $(\text{EM}(I^i))$.

3. To fine-tune $\mathcal{M}$, minimize the similarity loss $(\mathcal{L}_{\text{ssim}})$ between $\mathcal{E}(I^i)$ and $\text{EM}(I^i)$, such that $\mathcal{M}' = \underset{\mathcal{M}}{\text{argmin}} \mathcal{L}_{\text{ssim}}(\mathcal{E}(I^i), \text{EM}(I^i))$.

4. The SMN with $\mathcal{M}'$ predicts the final output as: $\mathbb{D}\Big(\mathbb{E}(I_i); [\sum_k^{\mathcal{C}} W_k^c \sum_{x,y} \mathbb{E}(I_i)] \cdot \mathcal{M}'\Big)$

It's important to note that during the training phase, $\mathcal{M}$ is trained, with each individual $\mathcal{M}$ being mapped to unique domains, representing different characteristics of contextual semantic information. Hence, $\mathcal{M}_i$ represents sub-domain $\mathcal{X}_i$, where the intersection of all $\mathcal{X}_i$ equates to the dataset $\mathcal{X}$, but no intersection exists among individual $\mathcal{X}_i$. During the inference phase, the similar $\mathcal{M}_i$ is derived by fine-tuning the Shape-Memory Network (SMN) for sample $x_i \in \mathcal{X}_i$. This process showcases how the SMN redeploys its saved structure by modifying its architecture, which led to the network being dubbed the *Shape-Memory Network*, and Fig. 5 in the manuscript verifies the effective utilization of similar $M_i$. Additionally, an illustration of the optimization of $\mathcal{M}$, as well as the provision of the final predicted segmentation map by the SMN, is presented in Algorithm 1.

---

**Algorithm 1:** Fine-tuning and Inference of the Shape-Memory Network

---

**Input** : sample $x_i$ in test-set ($\mathcal{X} \subset \mathbb{R}^{H \times W \times 3}$), such that $x_i \in \mathcal{X}$, where $H$ and $W$ are height and width, respectively, and the pre-trained SMN ($M$).

**Output** : Predicted segmentation map ($y_i \in \mathcal{Y} \subset \mathbb{R}^{H \times W \times C}$) corresponding to input ($x_i$), where $C$ is the number of category.

**Assumption** : $\mathcal{X} = \bigcup_i^N \mathcal{X}_i$ where $N$ is the number of subset. $\bigcap_i^N \mathcal{X}_i = \emptyset$, indicating that the $\mathcal{X}_i$ represents distinct domain.

$\bar{y}^i \leftarrow \mathbb{D}\Big(\mathbb{E}(I_i); [\sum_k^{\mathcal{C}} W_k^c \sum_{x,y} \mathbb{E}(I_i)] \cdot \mathcal{M}\Big)$ ;                    /* Predict pseudo-label */

$\bar{\mathcal{C}}_1 \leftarrow -\bar{y}^i \log \bar{y}^i$ ;                         /* Entropy-map by segmentation pipeline */

$\bar{\mathcal{C}}_2 \leftarrow - \sum_c^C \bar{d}^c(I^i)\|_{h,w} \log \bar{d}^c(I^i)\|_{h,w}$ ;                    /* Entropy-map by EM algorithm */

$\mathcal{M}' \leftarrow \underset{\mathcal{M}}{\text{argmin}} \mathcal{L}_{\text{ssim}}(\mathcal{E}(I^i), \text{EM}(I^i))$ ;                                /* Fine-tune $\mathcal{M}$ */

$y^i \leftarrow \mathbb{D}\Big(\mathbb{E}(I_i); [\sum_k^{\mathcal{C}} W_k^c \sum_{x,y} \mathbb{E}(I_i)] \cdot \mathcal{M}'\Big)$;

---

To summarize, the mechanism of the SMN is (1) to store the appropriate architecture for a certain domain; (2) to restore its structure corresponding to the input domain by fine-tuning a parameter; and (3) to provide precise prediction to the input image. Here, the structural mutation is achieved by fine-tuning the control signal that supervises the connections of neurons. The neural connections in the SMN are activated or deactivated based on the control signal, and thus fine-tuning the parameter that supervises the control signal enables the SMN to modify its network structure corresponding to the input image.

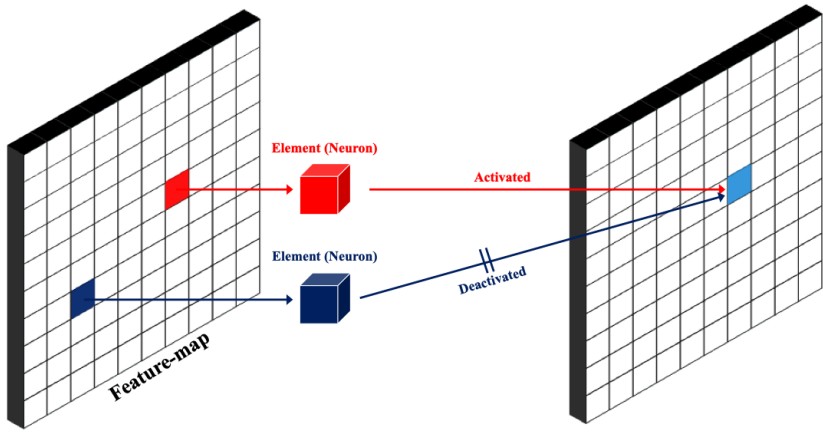

Appendix Figure 1: Schematic illustration of control neurons in the feature-maps.

Appendix Fig. 1 illustrates how the control signal achieves the structural mutation. By activating and deactivating the output of each neuron, which is the individual element in a feature-map, the condition signal changes the structure of the SMN, and thus the control signal supervises the adaptive domain adaptation with respect to the contextual semantic information of inputs. Thereby, the structural adjustment by the control signal fine-tuned with the contextual semantic information can bring out a superior segmentation performance of the SMN.

**Contributions** To summarize, our contributions, in this paper, are listed below:

- **Construction of Shape-Memory Network**. We designed the shape-memory network that can explicitly interpret the contextual semantic information by employing the run-time adaptation method via structural modification.

- **Design of Control Neuron**. We newly devised a control neuron that can adaptively change the connections to other control neurons, leading to the implementation of structural modification of the SMN. This mechanism represents the close emulation of the human brain connectome and synapse mechanism.

- **Implementation of Entropy-Map Reconstruction Algorithm**. For the explicit interpretation of the contextual semantic information in the SMN, we newly devised the entropy-map reconstruction algorithm to train the SMN using the class activation maps regarding the contextual semantic information. The devised algorithm incorporates the contextual semantic information in the training of the SMN.

## B    ENTROPY-MAP RECONSTRUCTION FROM CLASS ACTIVATION MAP

In the previous research (Zhou et al., 2016), it was revealed that the Class Activation Map (CAM) identifies the regions of significant relevance to the primary task. As a result, when tasked with density regression, the CAM is influenced to concentrate on areas specific to the target object ($c$), leading to the derivation of **Proposition V**. In this context, density refers to the proportion of pixels in the input image representing the target object compared to the total pixel count. In response, we developed the Shape-Memory Network, which integrates multi-label density regression tasks to yield multiple CAMs for each category ($c$). Parameters specific to the density regression process for each category are then multiplied with the encoded feature-map, and then, a CAM for each category is generated. Subsequently, we incorporate the CAMs that exhibit attention areas for each category to produce a pseudo-entropy-map 2.

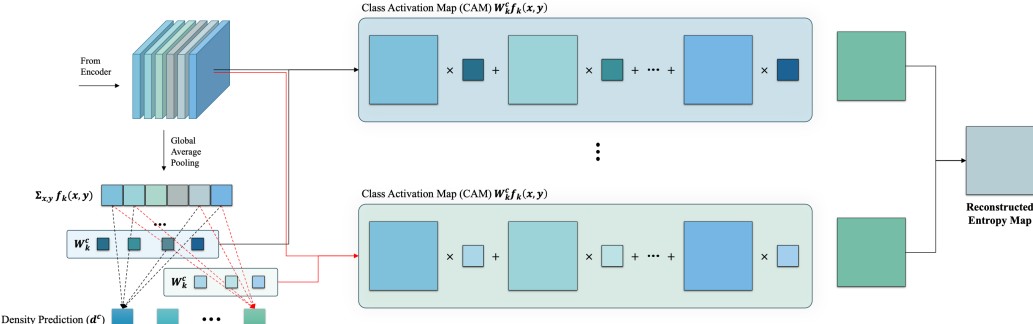

Appendix Figure 2: Schematic illustrations of reconstructing entropy-map from the class activation map (CAM). Each CMAs for each category are leveraged to generate entropy-map via a probability-based normalization method.

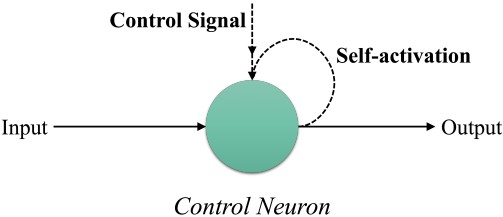

Appendix Figure 3: Schematic illustration of control neurons.

## C CONTROL NEURON

Note that the control neuron represents an element in a feature-map (See Appendix Fig.1). Therefore, the control neuron refers to pixel-wise activation rather than convolutional weights. Suppose there are $\mathbb{N}$ numbers of control neurons in the SMN, and each control neuron has individual intrinsic threshold value ($\lambda_n$) for $n^{\text{th}}$ control neuron. The output of the control neuron is activated when (1) the value of the control signal ($s^{\text{ct}}$) is above the intrinsic threshold value, such that $s^{\text{ct}} \geq \lambda_n$ or (2) self-activation is true. Therefore, the $s_n^{\text{out}}$ is activated when the following condition is satisfied:

$$(\text{self-activation})|(s^{\text{ct}} \geq \lambda_n) \tag{6}$$

Here, the self-activation indicates that the current control neuron (pixel or element) is more informative than other elements in the same feature-map. To avoid the loss of the informative features from the feature extraction process, the self-activation is designed. Therefore, the logical or ($|$) operator is placed in Eq. 6. Suppose $f_k$ for the encoded feature-map by $\mathbb{E}$. In $f_k$ with its height ($\mathcal{H}_k$) and width ($\mathcal{W}_k$), the number of control neurons are $\mathcal{H}_k\mathcal{W}_k$, and the $n^{\text{th}}$ control neuron is more informative when the condition below is satisfied:

$$s_n^{\text{out}} \text{ is in top} - k\% \text{ among all elements in } f_k. \tag{7}$$

In the previous study Lee et al. (2022), the sampling elements met the Eq. 7 is formulated as below:

$$s_n^{\text{out}} \text{ is informative when } s_n^{\text{out}} \geq m_{f_k} + z_k * \text{v}_{f_k} \tag{8}$$

where $m_{f_k}$ and $\text{v}_{f_k}$ are the mean and the standard deviation values of all elements in $f_k$, and the $z_k$ refers to a statistical z-value for the Z-table corresponding to $k\%$.

To generalize, suppose a feature-map ($\mathcal{F}$) in the SMN, and the $n^{\text{th}}$ control neuron in $\mathcal{F}$. Therefore, we can define the function ($g(s_n^{\text{out}}; \mathcal{F})$) that determines whether $s_n^{\text{out}}$ is informative or not as below:

$$g(s_n^{\text{out}}; \mathcal{F}) = \begin{cases} 1(\text{if } s_n^{\text{out}} \geq m_{\mathcal{F}} + z_k * \text{v}_{\mathcal{F}}) \\ 0(\text{else}) \end{cases} \tag{9}$$

In this case, the $z_k$ is not trainable since the $z_k$ is not arithmetically placed, but in the conditional statement. To make the $z_k$ be trainable, the Heaviside step function and its approximation are utilized. Additionally, the Heaviside step function ($H(x)$) is approximated to the sigmoid function ($\sigma(-2\alpha x)$) with a large value of $\alpha$. Therefore, we formulate the Eq. 9 as below:

$$\begin{aligned} g(s_n^{\text{out}}; \mathcal{F}) &= H(g(s_n^{\text{out}}; \mathcal{F}) - m_{\mathcal{F}} + z_k * \text{v}_{\mathcal{F}}) \\ &= \sigma\big(-2\alpha(g(s_n^{\text{out}}; \mathcal{F}) - m_{\mathcal{F}} + z_k * \text{v}_{\mathcal{F}})\big) \end{aligned} \tag{10}$$

Here, $\alpha$ and $z_k$ are trainable. Therefore, the self-activation that determines the current condition neuron is informative or not is trained during the training phase, and the pre-trained self-activation determines the activation of the condition neuron in the inference phase.

Furthermore, another condition in Eq. 6 related to the control signal is formulated using the approximation of the Heaviside step function as below:

$$\sigma\big(-2\alpha(s^{\mathrm{ct}} - \lambda_n)\big) \tag{11}$$

In addition, the logical or operator is replaced by the addition operator in arithmetic and analysis, and thus the Eq. 6 is substituted as below:

$$\sigma\big(-2\alpha(g(s_n^{\mathrm{out}}; \mathcal{F}) - m_{\mathcal{F}} + z_k * \mathrm{v}_{\mathcal{F}})\big) + \sigma\big(-2\alpha(s^{\mathrm{ct}} - \lambda_n)\big) \tag{12}$$

Therefore, let the input signals be $s^{\mathrm{in}}$, and thus the final output ($\mathbf{s}_n^{\mathrm{out}}$) value of the $n^{\mathrm{th}}$ control signal is provided as below:

$$\mathbf{s}_n^{\mathrm{out}} = s^{\mathrm{in}} * \Big(\sigma\big(-2\alpha_1(g(s_n^{\mathrm{out}}; \mathcal{F}) - m_{\mathcal{F}} + z_k * \mathrm{v}_{\mathcal{F}})\big) + \sigma\big(-2\alpha_2(s^{\mathrm{ct}} - \lambda_n)\big)\Big) \tag{13}$$

Here, in addition to the trainable parameters in **Appendix A**, the $z_k$, $\alpha_1$, and $\alpha_2$ are trainable, and $z_k$ represents the adaptive threshold to discriminate the informative features, and $\alpha_1$ and $\alpha_2$ are the conditional values for the approximation. In the empirical analysis and experiments, the value of $\alpha$ is trained at nearly 10.0.

## D  EXPERIMENTAL ENVIRONMENT DESCRIPTION

**Implementations**   The experiments were implemented in the Apple Macbook Pro with M1 Max and 64GB unified memories. Besides, we developed our neural network and the state-of-the-art deep learning models using Tensorflow (for ARM processor) version 2.9.0 (Abadi et al., 2016) for precise implementation. For the training, the batch size (Bottou, 2010) of the training was set to 32, and the Adam optimizer was utilized with the default values of all parameters (Kingma & Ba, 2014). Every parameter of the neural networks and the optimizer was initialized with the Gaussian distribution, of which the mean and the standard deviation values are 0.0 and 1.0.

**Comparative Models**   To demonstrate the segmentation performance of the Shape-Memory Network (SMN), four groups of deep learning models were utilized as shown in the followings; (1) Baseline models including the early vanilla models of U-NetRonneberger et al. (2015), and Seg-Former (Xie et al., 2021); (2) Multi-Path models for the segmentation task including LADDER-NET (Zhuang, 2018) and MPDNet (Bai & Zhou, 2020); (3) SotA models for the segmentation task, including InterImage (Wang et al., 2022b) and BeiT-3 (Wang et al., 2022c); (4) SotA models for the video object segmentation (VOS), including Xmem (Cheng & Schwing, 2022) and AOST (Yang et al., 2022). The baseline networks were compared to demonstrate the standard feasibility of the SMN for the segmentation task. While the SoTA models, used here, were utilized to exhibit superior segmentation performance of the SMN for the benchmark datasets of scene parsing and autonomous driving, including VOS. Here, the best-performing SotA models were selected by referring to *Kaggle* benchmark lists. Additionally, the multi-path models were employed to compare the SMN in terms of the ensemble models for Eq. (1) in the manuscript.

**Dataset**   In the experiments, five categories of distinct datasets were employed to evaluate the segmentation performance of the SMN compared to other baseline and SotA models; (1) Scene parsing benchmark using ADE20K (Zhou et al., 2017) and Youtube-VOS (Xu et al., 2018); (2) Autonomous driving using BDD100K (Yu et al., 2020); (3) Aerial image datasets of Inria (Maggiori et al., 2017b;a) and LoveDA (Wang et al., 2021); (4) Medical Imaging datasets using MRI for a brain tumor (Buda et al., 2019) and ultrasound dataset for breast cancer (Al-Dhabyani et al., 2019); (5) Synthetic images of GTA5 (Richter et al., 2016). To demonstrate the general feasibility of SMN for semantic segmentation, the datasets for scene understanding and autonomous driving datasets were utilized. In addition, since the density, which is a crucial feature for SMN, of objects is most important in the segmentation of aerial images, the benchmarks using aerial images were utilized. Furthermore, to evaluate the scalability of the SMN, the medical imaging datasets and benchmark for the synthetic images were employed. Note that since the density of the target object, especially the disease area,

is a significant key feature in medical imaging, the SMN could be expected to provide its superior segmentation performance in the medical imaging field. Furthermore, the precise segmentation performance of the SMN could provide the potential for transfer learning and extensibility to large-scale models. For training models, the images in each dataset are divided into ten-fold for the $k$-fold cross-validation.

**Comparison to Domain Adaptation Models**   A great deal of DA methods, such as adversarial training (Ganin et al., 2016), maximum mean discrepancy minimization (Tzeng et al., 2014), unsupervised DA (Ganin & Lempitsky, 2015), and self-ensembling (French et al., 2017), have demonstrated success in reducing discrepancies between distinct source and target domains. However, methods typically assume the availability of labeled source domain data and unlabeled target domain data during the training phase, a condition that may not hold in real-world scenarios. Test-time DA (TTDA) methods, in contrast, aim to refine models at the inference stage, by leveraging the test data distribution without explicit access to the labels. Techniques such as transductive parameter transfer (Shu et al., 2018), and test-time self-supervised learning (Azimi et al., 2022; Lee et al., 2021; Wang et al., 2022a) have been proposed to bridge the gap between the training and test data distributions.

Despite the promising results achieved by the aforementioned methods, they are primarily designed for addressing domain discrepancies between two or more distinct domains, rather than within a single domain. To apply the domain adaptation method, two significantly distinct domains should be identified. However, in our study, the key factor for the domain discrepancy is contextual semantic information, and the contextual semantic information could be identified by the deep learning models, not by the human, and thus the labels for the different domains regarding the contextual semantic information could not be provided. Therefore, despite the promising performance of the domain adaptation decreasing the domain gap, the domain adaptation method could not be applied and implemented to resolve the issues addressed in the problem statement (Sec. 2).

## E   EXPERIMENTS

**Verification of $\mathcal{M}$**   Appendix Fig. 4 illustrates the similarity between $\mathcal{M}_i$ and $\mathcal{M}_j$ for samples of $x_i$ and $x_j$ in the same domain $\mathcal{X}$ alongside three error rates. This experiment was conducted to measure the justification that fine-tuning $\mathcal{M}$ could represent similar or different architecture for the SMN within different domains.

To measure the similarity, the following function is devised:

$$\mathcal{S}(x,y;r) := \begin{cases} 1 \ (\text{if } \frac{|x-y|}{x} \leq r) \\ 0 \ (\text{else}) \end{cases} \tag{14}$$

where $0 \leq r \leq 1$ represents the error rate, and thus $\mathcal{S}$ represent 1 if two elements is within the error rate. Here, the Similarity of the Intrinsic Threshold is calculated below:

Appendix Table 1: Detailed description of the datasets. To validate, 10-fold cross-validation was used.

| Dataset | Samples | Train | Test | Category |
|---------|---------|-------|------|----------|
| ADE20K | 27,574 | 24,817 | 2,757 | 150 |
| Youtube-VOS | 7,945 | 7,150 | 795 | 65 |
| BDD100K | 8,000 | 7,200 | 800 | 20 |
| Inria | 144,000 | 129,600 | 14,400 | 2 |
| LoveDA | 4191 | 3,772 | 419 | 8 |
| BrainMRI | 7,858 | 7,073 | 785 | 2 |
| BUSI | 789 | 709 | 80 | 2 |
| GTA5 | 24,966 | 22,470 | 2,496 | 27 |

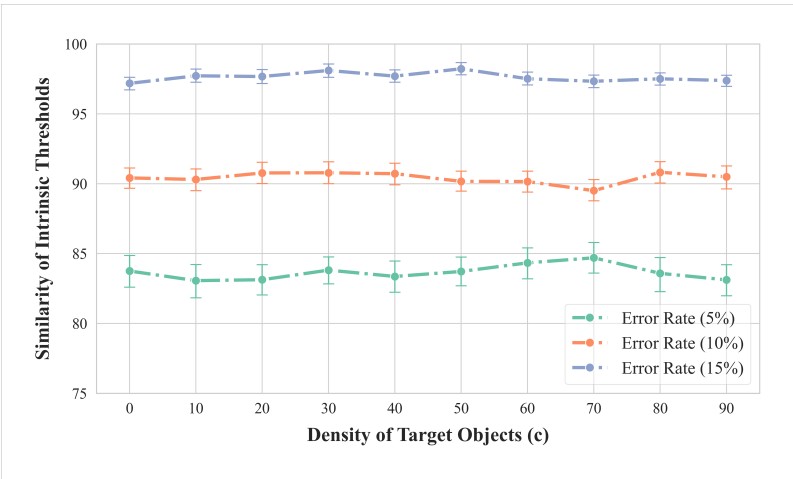

Appendix Figure 4: Similarity of the intrinsic threshold of control neurons containing the similar density of the target objects.

$$\frac{1}{C\mathbb{N}} \sum_{c,n}^{C,\mathbb{N}} \mathcal{S}(\mathcal{M}_i\|_{c,n}, \mathcal{M}_j\|_{c,n}; r) \tag{15}$$

If the error rate decreases, the Similarity of the Intrinsic Threshold guarantees a higher similarity, whereas a large value of the error rate is a rough condition. Therefore, Appendix Fig. 4 verifies that the fine-tuned $\mathcal{M}$ exhibits similar values regarding the same domain.

The SMN contains a small number of parameters compared to other state-of-the-art models, but the SMN significantly provides precise prediction in the segmentation task due to its effective fine-tuning mechanism. Additionally, despite the fine-tuning mechanism of the SMN, the SMN exhibits an efficient FPS due to only a small number of parameters ($\mathcal{M}$) being optimized in an inference phase.

**Segmentation Performance**  This section illustrates the evaluation results of our model compared to other deep learning models, including baseline models, multi-path models, SotA models for segmentation, and the SotA models for VOS.

In addition, the figure below illustrates the samples of the predicted segmentation by the SMN and other comparative models in eight datasets.

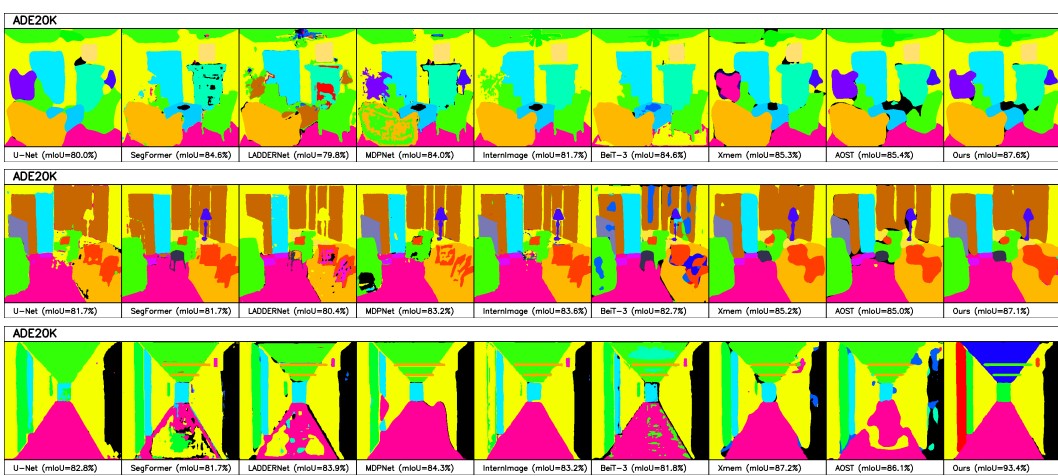

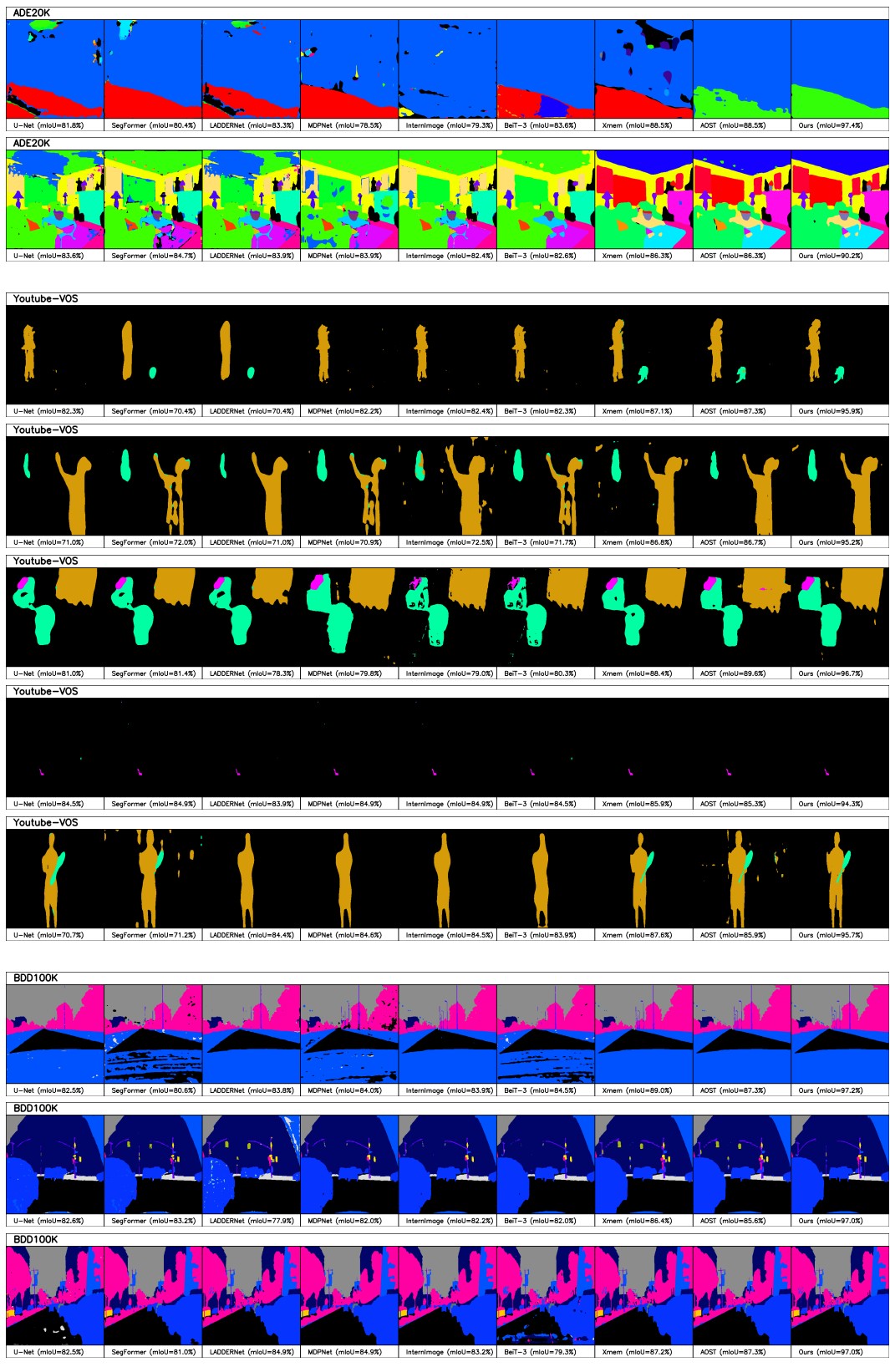

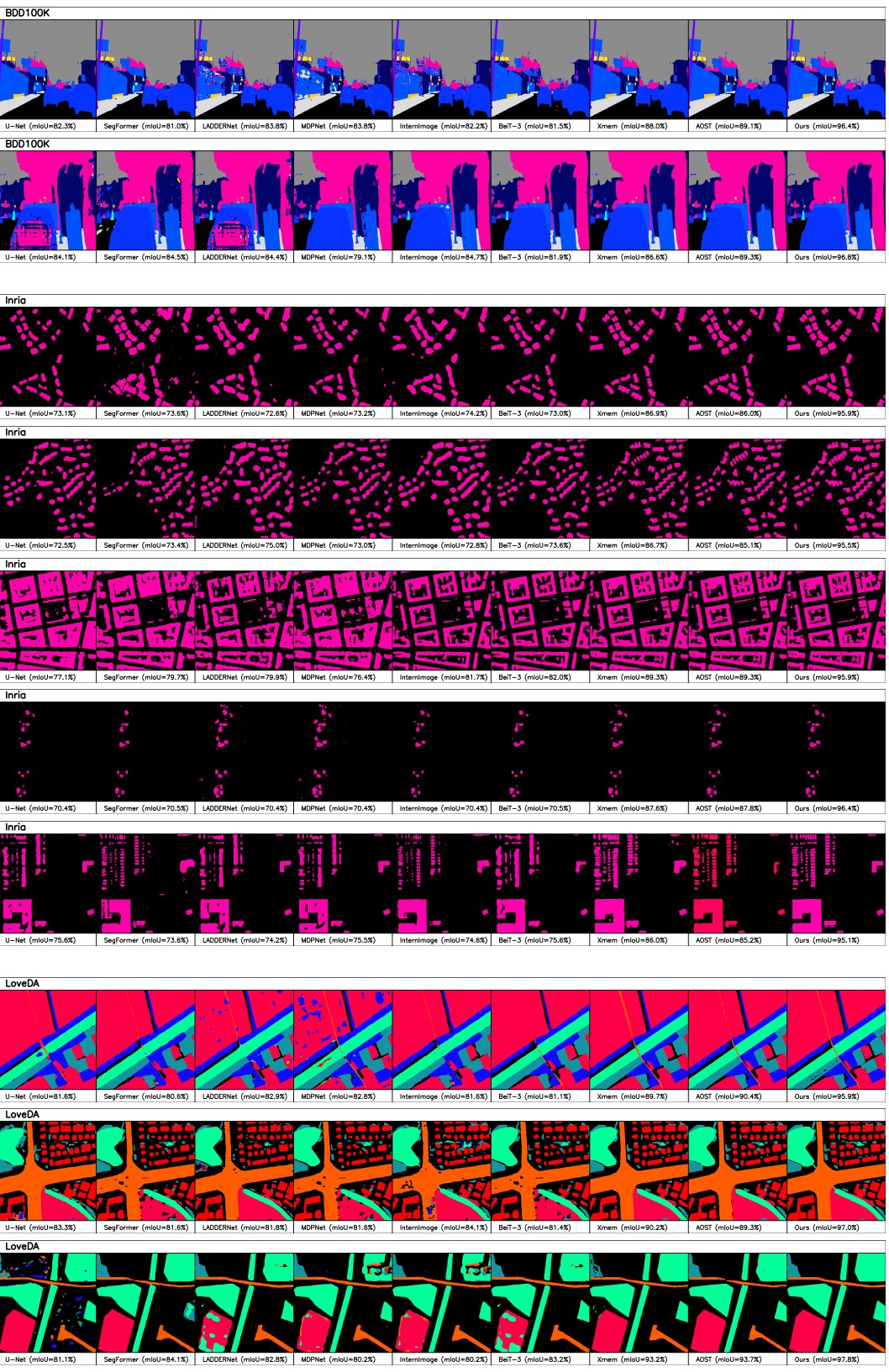

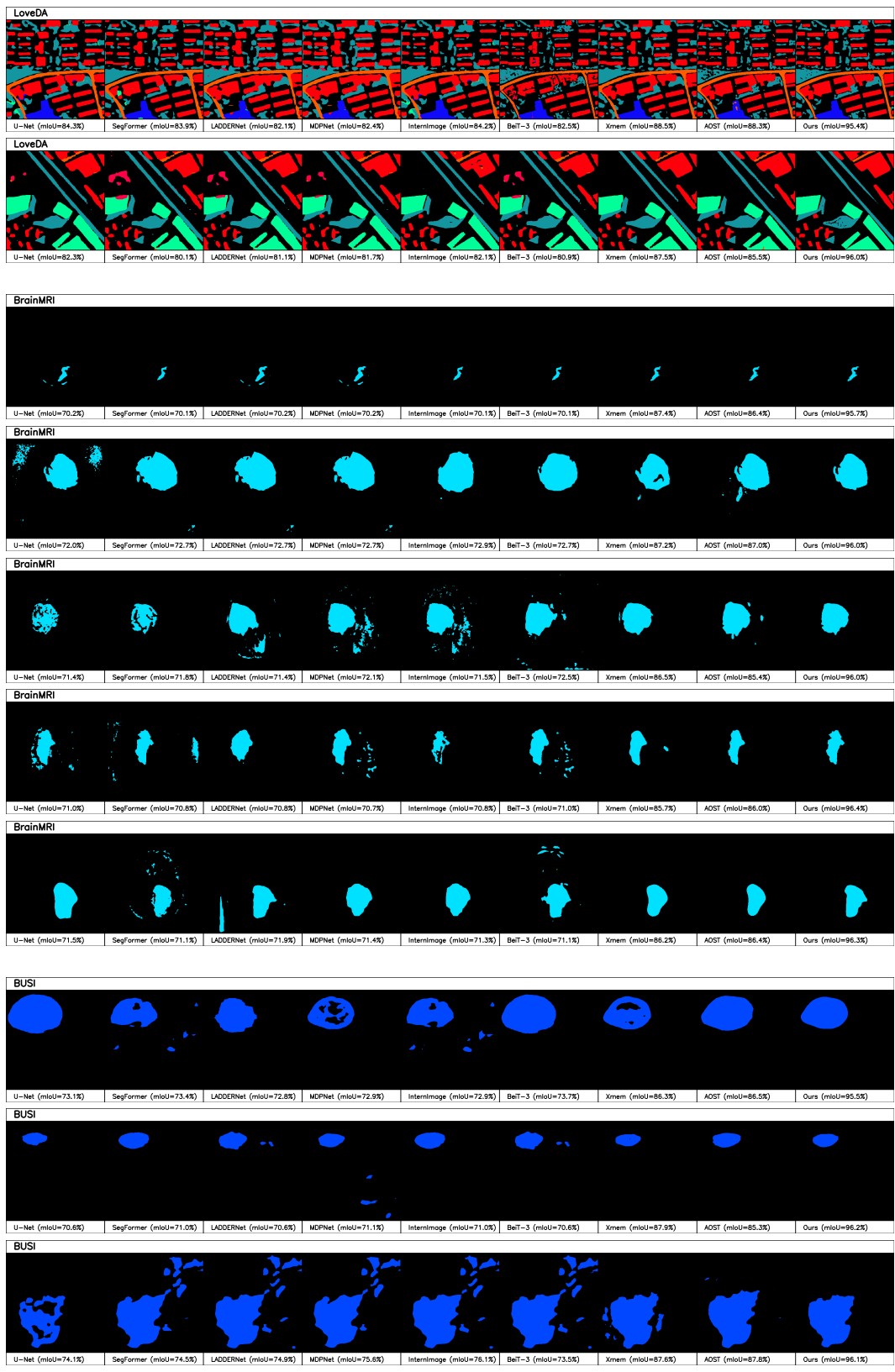

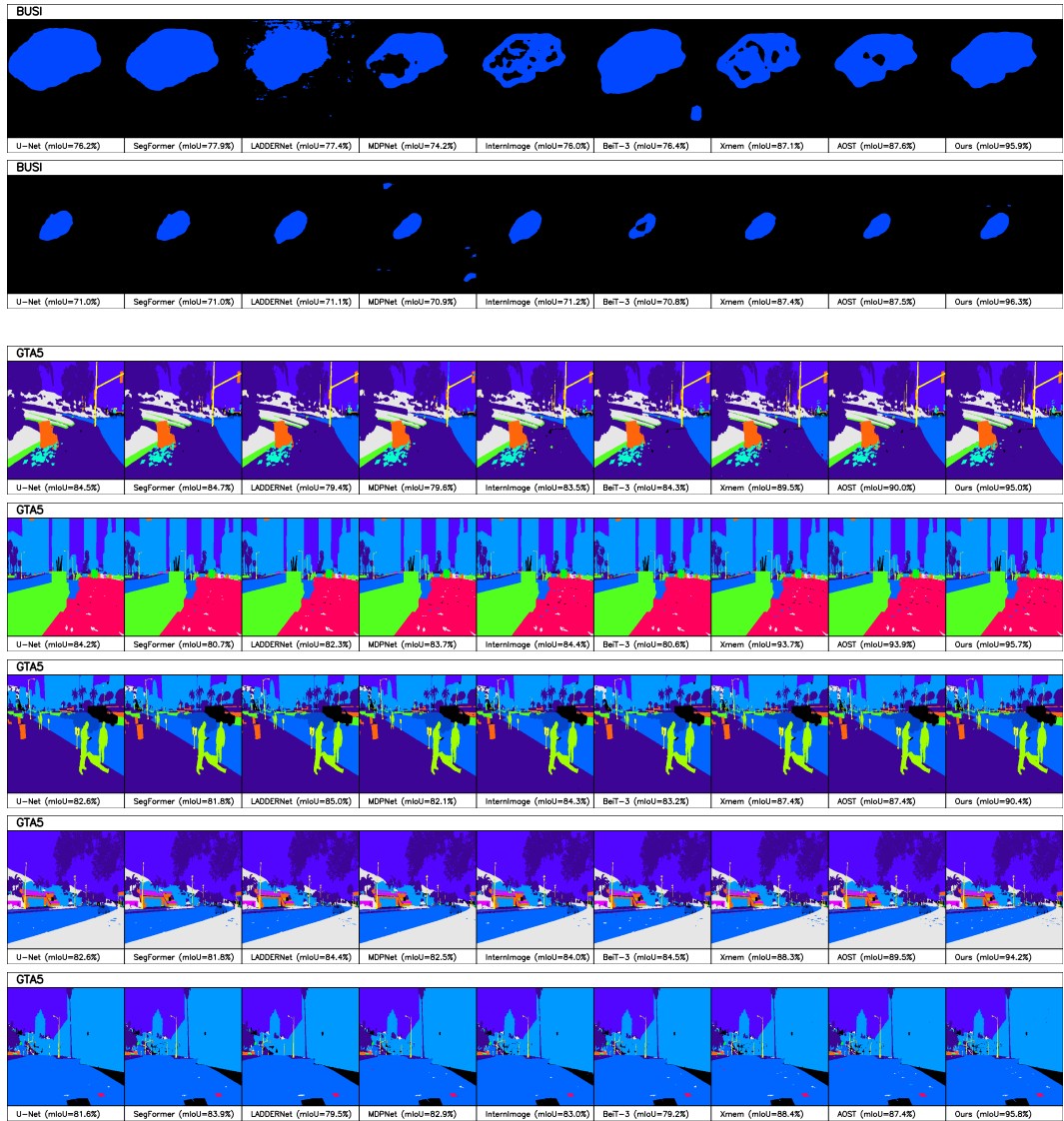

Appendix Table 3: Segmentation Results of the SMN and other comparative models.

**Network Complexity** The SMN contains the trainable parameters of (1) parameters of encoder and decoder, such that $\theta_{\mathbb{E}}$ and $\theta_{\mathbb{D}}$; (2) parameters $(W_k^c)$ for the CAM pipeline as a dense layer; (3) matrix $(\mathcal{M})$ to map the predicted density to a control signal. To verify the feasibility and scalability for a real-world application, we compared the SMN to other deep learning models in terms of the number of parameters (# of Param), the FLoating point Operations Per Second (FLOPs), and Frame Per Second for generating predictions (FPS). The predictions were performed using the Apple Macbook Pro with M1 Max and 64GB unified memories.

Appendix Table 2: Comparison analysis in terms of mean IoU.

|  |  | ADE20K | Youtube-VOS | BDD100K | GTA5 |
|---|---|---|---|---|---|
| 2*Baseline Model | U-Net | 42.61% (±3.75) | 77.12% (±3.38) | 36.69% (±1.41) | 65.84% (±1.81) |
|  | SegFormer | 46.72% (±3.93) | 81.07% (±2.37) | 42.59% (±3.83) | 65.99% (±2.17) |
| 2*Multi-Path | LADDERNet | 52.66% (±3.44) | 86.04% (±3.53) | 41.13% (±3.44) | 68.64% (±3.61) |
|  | MPDNet | 44.3% (±3.11) | 83.57% (±2.54) | 40.03% (±3.6) | 70.69% (±3.42) |
| 2*Seg SotA | InternImage | 51.04% (±2.67) | 85.13% (±2.2) | 47.25% (±3.5) | 70.94% (±3.27) |
|  | BEiT-3 | 51.67% (±2.08) | 86.67% (±3.02) | 39.85% (±1.22) | 70.9% (±3.23) |
| 2*VOS | Xmem | 44.88% (±1.78) | 83.36% (±3.31) | 42.69% (±3.35) | 68% (±2.31) |
|  | AOST | 52.06% (±3.12) | 87.09% (±2.04) | 43.09% (±1.41) | 71.06% (±2.32) |
| 2*Ours | Ours - SA | 48.84% (±2.42) | 85.66% (±1.04) | 43.81% (±3.59) | 69.07% (±2.97) |
|  | Ours | **55.76% (±2.9)** | **88.66% (±2.44)** | **48.83% (±1.14)** | **76.58% (±1.14)** |
|  |  | Inria | LoveDA | BrainMRI | BUSI |
| 2*Baseline Model | U-Net | 62.96% (±3.34) | 47.71% (±3.12) | 75.11% (±1.64) | 63.69% (±2.55) |
|  | SegFormer | 67.97% (±3.19) | 51.33% (±3.34) | 74.28% (±3.36) | 71.41% (±3.19) |
| 2*Multi-Path | LADDERNet | 64.77% (±3.09) | 49.66% (±3.59) | 69.75% (±3.34) | 60.36% (±2.88) |
|  | MPDNet | 64.51% (±1.09) | 48.25% (±3.43) | 67.43% (±3.89) | 67.51% (±1.28) |
| 2*Seg SotA | InternImage | 68.6% (±3.27) | 49.81% (±1.17) | **76.08% (±3.82)** | 70.47% (±2.27) |
|  | BEiT-3 | 66.69% (±1.82) | 49.63% (±1.75) | 66.08% (±3.78) | 67.46% (±1.63) |
| 2*VOS | Xmem | 64.85% (±2.78) | 51.29% (±3.12) | 61.57% (±3.04) | 67.24% (±3.41) |
|  | AOST | 69.3% (±3.75) | 50.57% (±3.43) | 75.22% (±2.83) | 60.16% (±3.2) |
| 2*Ours | Ours - SA | 68.6% (±2.57) | 50.4% (±2.69) | 68.86% (±2.95) | 72.99% (±3.42) |
|  | Ours | **72.72% (±1.06)** | **54.28% (±1.31)** | 74.82% (±1.77) | **75.22% (±1.24)** |

Appendix Table 4: Comparison analysis in terms of networks' complexities.

|  | U-Net | SegFormer | InternImage | BeiT-3 | Xmem | AOST | SMN (Ours) |
|---|---|---|---|---|---|---|---|
| Resolution | $512 \times 512$ | $512 \times 512$ | $384 \times 384$ | $384 \times 384$ | $512 \times 512$ | $512 \times 512$ | $512 \times 512$ |
| # of Param | 31.0M | 64.1M | 335M | 1843M | - | 65.6M | 47.5M |
| FLOPs | 224.6G | 95.7G | 163.2G | 2859.9G | - | - | 549.8G |
| FPS | 42.5 | 30.7 | 42.6 | 10.2 | 41.7 | 35.2 | 32.8 |

## F    DISCUSSION

**Extension to Other Tasks**    Our framework demonstrates significant potential for extension beyond semantic segmentation tasks. As illustrated in Appendix Fig. 5, the SMN architecture can be generalized via a modular design approach: maintaining the encoder with control neurons while allowing customization of the header and pretext task for specific applications. The adaptability of the SMN is achieved by two key components: (1) the latent features extracted from the encoder and (2) the control signals derived from the features. The latent features, representing high-level semantic information, are processed through task-specific headers to generate appropriate outputs (e.g., class probabilities for classification, bounding box coordinates for detection), while the control signals guide the structural adaptation of the network based on a pretext task appropriate for the target application. While our segmentation implementation uses density-based pretext tasks to identify spatial information, other applications might employ different self-supervised learning objectives - for instance, classification tasks could utilize feature correlation learning based on variational auto-encoder, while detection tasks might benefit from pretext tasks using object localization patterns.

The adaptability of our SMN architecture extends beyond semantic segmentation tasks through its modular design approach, as illustrated in Fig. 5. The architecture maintains its main feature extraction mechanism with control neurons while enabling task-specific customization through two key components: the decoder (header) and the pretext task. This design principle allows the network to be adapted for various computer vision tasks while preserving the benefits of our control neuron mechanism.

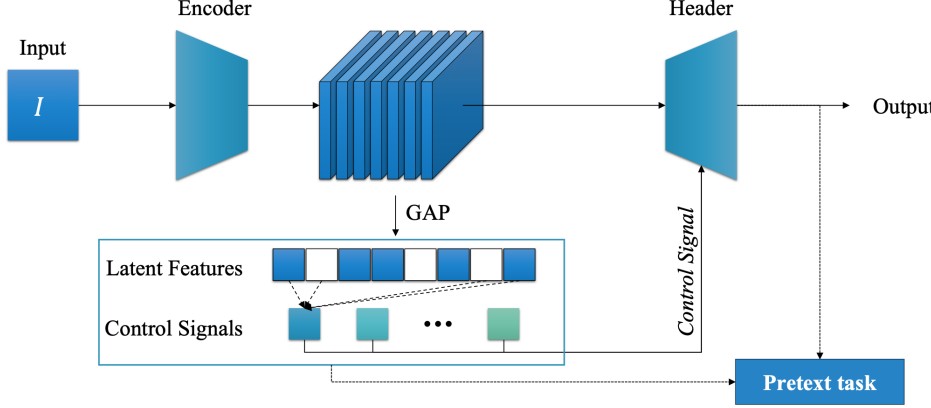

Appendix Figure 5: Generalized pipeline of SMN for various computer vision tasks.

In our preliminary study, we demonstrate this adaptability in object detection tasks, where we modified only the decoder while maintaining the control neuron mechanism and semantic information-based optimization. Our initial experiments on the COCO dataset show competitive results compared to recent state-of-the-art detection models like DiffusionDet, achieving 47.43 AP, 65.64 $AP_{50}$, and 52.21 $AP_{75}$. Notably, our approach achieves this performance with minimal architectural modifications, demonstrating particularly strong results in $AP_l$ (63.24) for large object detection. For detection tasks, we leverage the same semantic information optimization strategy as used in segmentation, demonstrating the transferability of our core mechanism across different vision tasks.

| COCO | AP | $AP_{50}$ | $AP_{75}$ | $AP_s$ | $AP_m$ | $AP_l$ |
|---|---|---|---|---|---|---|
| DiffusionDet (1@ 500) | 47.18 | 65.74 | 51.42 | 31.18 | 50.19 | 62.24 |
| DiffusionDet (4@ 500) | 47.36 | 65.62 | 52.13 | 30.72 | 50.37 | 63.18 |
| Ours (SMN) | 47.43 | 65.64 | 52.21 | 30.8 | 50.39 | 63.24 |

The flexibility of the SMN extends further through our generalized pipeline (Fig. 5), where the pretext task can be customized for different applications. While segmentation and detection tasks benefit

from semantic information optimization, other computer vision applications may require different self-supervised learning approaches. For classification tasks, we are exploring various pretext task approaches leveraging auto-encoders and VAE architectures, including feature correlation learning between augmented views, rotation prediction, and solving jigsaw puzzles of image patches. This modular architecture design, separating the core feature extraction mechanism from task-specific components, ensures that the primary strengths of our approach remain effective across different applications. The empirical experiments in both segmentation and detection tasks and additional ongoing exploration in classification demonstrate the broader applicability of our brain-inspired approach across various computer vision tasks, with the selection of appropriate pretext tasks being the key consideration for each specific application.

**Reproducibility** More detailed experimental results, including class-wise IoU values, and the code for the SMN will be available at https://github.com/Anonymous/Repo.

