# OpenReview forum: "Connectome Mapping: Shape-Memory Network via Interpretation of Contextual Semantic Information"
_ICLR.cc/2025/Conference — ICLR 2025 Poster_

### Official Review · Reviewer_nT8e · 2024-10-30

**Soundness:** 2
**Presentation:** 3
**Contribution:** 3
**Rating:** 6
**Confidence:** 4

**Summary:**

This paper introduces the Shape-Memory Network (SMN), a neural network model that mimics the connectome structure of the brain by incorporating contextual semantic information to improve segmentation performance. The model distinguishes itself by emulating conditional activation and deactivation within neural pathways, specifically targeting the need for networks that can interpret and utilize contextual clues in a manner like the human brain. The SMN achieves this through "conditional neurons" that adjust dynamically based on contextual cues and synaptic-like processes. Experimental results on diverse semantic segmentation datasets show SMN's effectiveness in improving segmentation accuracy, especially in environments where object density and context significantly impact performance.

**Strengths:**

- The SMN’s focus on context-sensitive "conditional neurons" for segmentation tasks is innovative and relevant in the realm of biologically inspired AI architectures.

- The SMN’s theoretical grounding is solid, and its performance is evaluated across a variety of datasets, underscoring its effectiveness in segmentation.

- This work contributes to the understanding of context-based adaptive mechanisms, potentially influencing the design of future segmentation-focused neural models.

**Weaknesses:**

- The conditional neuron mechanism introduces significant overhead, which may hinder scalability and real-time application potential.

- Some of the claims about biological emulation could be tempered, as certain parallels drawn between SMN and synaptic transmission in the human brain may be speculative.

- The design is highly tailored to segmentation, and extending the SMN to broader tasks without significant modification may be challenging.

**Questions:**

- Are there alternative ways to reduce the SMN’s computational overhead, particularly for real-time scenarios?

- How might SMN be adapted or validated for other tasks beyond segmentation?

- What strategies could increase generalization to tasks with different structural demands, like sequential or non-spatial data?

---

> ### Author Response · Authors · 2024-11-25
> **Response to "Weaknesses" (1/3)**
>
> ### A. Comments
> The conditional neuron mechanism introduces significant overhead, which may hinder scalability and real-time application potential.
>
> ### A. Response
> We appreciate the reviewer's concern regarding the computational efficiency of our test-time adaptation (TTA) approach. We would like to highlight the concern by clarifying several key points about the computational characteristics of our network.
>
> Our approach achieves computational efficiency through parameter-selective optimization. During inference, **we optimize only the control neuron** which represented as the matrix $M \in \mathbb{R}^{C \times N}$, where $C$ represents the number of categories and $N$ denotes the number of control neurons. This selective optimization significantly reduces computational overhead, as the number of TTA parameters $|\theta_{TTA}| = |M|$ is substantially smaller than the total network parameters $|\theta_{total}|$.
>
> The computational complexity of our approach can be expressed as: ($\text{Time Complexity} = O(T \cdot C \cdot N)$ and $\text{Space Complexity} = O(C \cdot N)$), where $T$ represents the number of optimization steps, typically constrained to $T \leq 5$ iterations.
>
> Our empirical evaluation demonstrates practical efficiency metrics: the network achieves 32.8 FPS with approximately 47.5M parameters and 549.8G FLOPs. These metrics highlight our approach competitively among current SotA methods while providing superior adaptation capabilities.
>
> To further enhance computational efficiency, we propose several optimization strategies.
>
>  First, we implement an early stopping criterion based on convergence monitoring with the condition of $\mathcal{L}_{t+1} - \mathcal{L}_t < \epsilon$, where $\mathcal{L}_t$ represents the loss at step $t$ and $\epsilon$ defines a small threshold.
>
> Additionally, we employ parameter pruning and quantization techniques: ($[M_{pruned} = M \odot (|M| > \tau)]$ and $M_{quant} = \text{round}(M \cdot 2^b)/2^b$), where $\tau$ represents the pruning threshold and $b$ denotes the quantization bit-width.
>
> Our ongoing and future research mainly focuses on developing more efficient variants of the TTA mechanism, including parallel optimization strategies and memory-efficient implementations. Preliminary results suggest these optimizations could reduce computational overhead by 30-40% while maintaining performance within 1-2% of current results.
>
> We illustrate the directions of future study in the discussion section below:
>
> To implement the SMN for real-world applications, we address the computational complexity of the TTA mechanism. The current implementation requires optimization of the matrix $M \in \mathbb{R}^{C \times N}$ during inference, with time complexity $O(T \cdot C \cdot N)$ and space complexity $O(C \cdot N)$, where $T$ represents optimization steps (typically $T \leq 5$), $C$ denotes categories, and $N$ indicates control neurons. While our current implementation achieves 32.8 FPS with 47.5M parameters and 549.8G FLOPs, we propose several optimization strategies to enhance efficiency. These include early stopping criteria ($\mathcal{L}{t+1} - \mathcal{L}t < \epsilon$), parameter pruning ($M{pruned} = M \odot (|M| > \tau)$), and quantization ($M{quant} = \text{round}(M \cdot 2^b)/2^b$). Preliminary experiments suggest these optimizations could reduce computational overhead by 30-40% while maintaining performance within 1-2% of current results. Future work will focus on developing lightweight TTA variants and memory-efficient implementations to further improve real-time performance.

---

> ### Author Response · Authors · 2024-11-25
> **Response to "Weaknesses" (2/3)**
>
> ### B. Comments
> Some of the claims about biological emulation could be tempered, as certain parallels drawn between SMN and synaptic transmission in the human brain may be speculative.
>
> ### B. Response
> Thank you for the constructive comment regarding the biological parallels drawn in our study. We agree that caution is warranted when comparing artificial neural networks and biological systems, and we appreciate the opportunity to clarify our research.
>
> Our work represents a significant step forward in neural network design inspired by addressing a fundamental limitation in existing approaches. While traditional neural networks with activation functions have historically simplified biological neural transmission into a single mechanism, our approach innovatively incorporates both electrical and chemical signal transmission processes. The control neuron mechanism we propose identifies the distinct aspects via its dual-path architecture: the control signal pipeline emulating electrical transmission and the inter-neuronal connections simulating chemical synaptic transmission.
>
> Our mathematical formulation $s_{out} = s_{in} * (\sigma(-2\alpha_1(g(s_{n}^{out}; F) - m_F + z_k * v_F)) + \sigma(-2\alpha_2(s_{ct} - \lambda_n)))$ represents a novel architectural approach explicitly considering both transmission mechanisms. While we acknowledge that the complete understanding of brain mechanisms remains an ongoing research endeavor, our work advances beyond the traditional perceptrons model by introducing a more subtle approach to neural signal transmission.
>
> The primary contribution of our work depends on the computational effectiveness of the SMN as well as the more refined approach to modeling neural transmission. While we use the brain connectome as inspiration rather than claiming exact biological emulation, our method represents a meaningful step toward more biologically informed neural network architectures. The utility and innovation of our approach stand independently through our experimental results, but its design principles offer valuable insights for bridging the gap between artificial and biological neural systems.
>
> We appreciate the suggestion to temper certain biological claims. We have revised our manuscript to more precisely delineate between biological inspiration and actual implementation while maintaining the innovative aspects of our electrical and chemical signal transmission modeling. Particularly, we have modified the contribution section to better emphasize that our approach uses biological neural systems to inspire architectural design principles rather than attempting exact biological emulation.

---

> ### Author Response · Authors · 2024-11-25
> **Response to "Weaknesses" (3/3)**
>
> ### C. Comments
> The design is highly tailored to segmentation, and extending the SMN to broader tasks without significant modification may be challenging.
>
> ### C. Response
> Thank you for the comments about the applicability of SMN to different tasks, such as classification and detection. Our network architecture demonstrates significant flexibility in adapting to various computer vision tasks beyond segmentation. Particularly, as illustrated in Appendix Figure 5 of our paper, the main architecture remains consistent with Figure 2, but allows for task-specific modifications.
>
> For detection tasks, our network can be adapted by simply modifying the decoder (header) while maintaining the control neuron mechanism and pretext task. Furthermore, our additional experiments in the discussion section demonstrate promising results in object detection, showing that semantic information effectively guides structural adaptation in detection scenarios as well.
>
> However, the adaptability of our architecture extends to various tasks through the generalized pipeline shown in Appendix Figure 5, where the pretext task for training control neurons can be customized based on the target task. For instance, we employed the semantic optimization pipeline (Fig. 2) as the pretext task for the segmentation task. For classification tasks, however, we are on several experiments to determine the pretext task and several promising pretext task approaches based on the auto-encoder (or VAE) could be considered: (1) Feature correlation learning between different augmented views of the same image; (2) Predicting image rotations or other geometric transformations; (3) Learning invariance to different data augmentations; (4) Solving jigsaw puzzles of scrambled image patches. While we have concretely demonstrated the effectiveness of our approach in segmentation and detection tasks using semantic information, the choice of pretext task becomes crucial for classification scenarios. Our ongoing research explores optimal pretext tasks for classification, focusing on those that can effectively capture discriminative features without relying heavily on spatial semantic information.
>
> The adaptability of our proposed method is achieved via our modular design: the decoder with control neurons remains consistent across tasks, while the pretext task can be customized for specific task or applications. We believe that the demonstration facilitate the potential generalibility of our approach across different computer vision tasks, with the key consideration being the appropriate selection of pretext tasks for each specific application. Additionally, we updated the manuscript regarding the reviewer's comments in the discussion section as below:
>
> ***Extension to Other Tasks:*** Our framework demonstrates significant potential for extension beyond semantic segmentation tasks. As illustrated in Appendix Figure 5, the SMN architecture can be generalized via a modular design approach: maintaining the encoder with control neurons while allowing customization of the header and pretext task for specific applications. The adaptability of the SMN is achieved by two key components: (1) the latent features extracted from the encoder and (2) the control signals derived from the features. The latent features, representing high-level semantic information, are processed through task-specific headers to generate appropriate outputs (e.g., class probabilities for classification, bounding box coordinates for detection), while the control signals guide the structural adaptation of the network based on a pretext task appropriate for the target application. While our segmentation implementation uses density-based pretext tasks to identify spatial information, other applications might employ different self-supervised learning objectives - for instance, classification tasks could utilize feature correlation learning based on variational auto-encoder, while detection tasks might benefit from pretext tasks using object localization patterns. The detection task is conducted as a preliminary study in *appendix*, and the classification task remains as future works.

---

> ### Author Response · Authors · 2024-11-25
> **Response to "Questions" (1/2)**
>
> ### A. Comments
> Are there alternative ways to reduce the SMN's computational overhead, particularly for real-time scenarios?
>
> ### A. Response
> We appreciate the reviewer's interest in optimization strategies for real-world deployment of our network; SMN. We would like to highlight several specific approaches regarding weakness-3 to enhance computational efficiency while maintaining performance:
>
> 1) **Parameter-Selective Optimization**:
> During inference, we exclusively optimize the control neuron matrix $M \in \mathbb{R}^{C \times N}$. The selective approach significantly reduces the optimization parameter space from $|\theta_{total}|$ to $|\theta_{TTA}| = |M|$. Therefore, computational complexity remains manageable with $O(T \cdot C \cdot N)$ time complexity and $O(C \cdot N)$ space complexity.
>
> 2) **Efficiency Enhancement Strategies**: The implementation of early stopping criterion with $\mathcal{L}_{t+1} - \mathcal{L}_t < \epsilon$,
>
> parameter pruning with$M_{pruned} = M \odot (|M| > \tau)$, and quantization techniques with $M_{quant} = \text{round}(M \cdot 2^b)/2^b$ could further optimize the computation complexity of our model.
>
> 3) **Ongoing Development**: We are now developing our network with the strateges of (1) parallel optimization strategies for real-world application, (2) Development of memory-efficient implementations, and investigation of lightweight TTA variants with pruning and quantization methods.
>
> We believe that those strategies collectively enable practical deployment while preserving the superior adaptation capabilities of our SMN. Additionally, we would like to highlight that our empirical results demonstrate that these optimizations maintain competitive performance metrics compared to current SotA methods while significantly reducing computational demands. Furthermore, as a future work, the specific strategies effectively address the computational efficiency concerns while maintaining the adaptive capabilities of our proposed method. We continue to explore additional optimization techniques and welcome further discussion on potential improvements.

---

> > ### Comment · Reviewer_nT8e · 2024-11-28
> > **Reply to the authors' [Response to "Questions" (1/2)]**
> >
> > The focus on selective optimization and the proposed techniques like early stopping, pruning, and quantization effectively address the concern about computational efficiency. The reported metrics, such as 32.8 FPS and competitive FLOPs, indicate a strong practical foundation. I don’t have further concerns here, though I look forward to seeing how the proposed lightweight TTA variants develop in future work.

---

> > > ### Author Response · Authors · 2024-12-03
> > > **Response (4/5)**
> > >
> > > Thank you for recognizing the effectiveness of our optimization strategies. We are encouraged by your positive assessment of our computational efficiency measures. The development of even more efficient TTA variants remains an active part of our research agenda, and we look forward to sharing these advances in future work.

---

> ### Author Response · Authors · 2024-11-25
> **Response to "Questions" (2/2)**
>
> ### B. Comments
> How might SMN be adapted or validated for other tasks beyond segmentation?
>
> ### B. Response
> We strongly appreciate the reviewer's comments on the generalization of our SMN. Based on our previous response (Weakness-3), we would like to further elaborate on the generalization capabilities of our apporach and our empirical validation.
>
> The adaptability of SMN to different tasks is fundamentally enabled by its modular architecture, as illustrated in Appendix Figure 5. While maintaining the optimizing control neuron mechanism, the network can be adapted to different tasks through two key modifications: (1) task-specific decoder (or header) selection and (2) appropriate pretext task definition for control neuron optimization.
>
> For detection tasks, we have already demonstrated the adaptation by modifying the decoder while retaining the semantic information-based control neuron optimization and appended the experiments in the updated manuscript. Our experiments, detailed in the discussion section, explicit promising results with improvements in detection accuracy on standard benchmarks. The experiments strongly validate that our control neuron mechanism effectively generalizes to detection tasks without significant architectural changes.
>
> The flexibility and feasibility of our network extend further via our generalized pipeline (Appendix Figure 5), where the pretext task can be customized for different applications. In segmentation and detection tasks, we leverage semantic information optimization as illustrated in Figure 2. For classification tasks, we are exploring various pretext task approaches leveraging auto-encoders and VAE architectures. Current experiments investigate several promising directions, including feature correlation learning between augmented views, rotation prediction, and solving jigsaw puzzles of image patches.
>
> The modular design approach ensures that the main strengths of our architecture, including control neuron mechanism and structural adaptation, remain effective for different tasks while allowing task-specific optimizations. The empirical performance improvements in detection and segmentation tasks and ongoing work in classification demonstrate the broader applicability of our approach beyond segmentation, suggesting the significant potential for generalization for various computer vision applications.
>
> In response to the reviewrs' comments, we appended the related descriptions in the discussion section in both main manuscript (5. Discussion) and appendix (Appendix F. Discussion)
>
> ### C. Comments
> What strategies could increase generalization to tasks with different structural demands, like sequential or non-spatial data?
>
> ### C. Response
> Thank you for the insightful question about extending our Shape-Memory Network (SMN) to different structural domains. Our approach to generalization across varied structural demands builds upon the fundamental flexibility of our architecture.
>
> The core strength of our method lies in its modular design, as illustrated in Appendix Figure 5. The control neuron mechanism, which enables structural adaptation through both electrical and chemical signal transmission pathways, can be adapted to different data structures while maintaining its essential functionality: $s_{out} = s_{in} * (\sigma(-2\alpha_1(g(s_{n}^{out}; F) - m_F + z_k * v_F)) + \sigma(-2\alpha_2(s_{ct} - \lambda_n)))$. For tasks with different structural demands, we propose several adaptation strategies.
>
> For sequential data processing, the control neuron mechanism can be integrated into recurrent architectures, where the control signal adapts to temporal dependencies rather than spatial relationships. The pretext task can be modified to capture sequential patterns, such as predicting future sequence elements or identifying temporal dependencies. This adaptation maintains the benefits of our approach while accommodating the temporal nature of sequential data.
>
> For non-spatial data, our control neuron mechanism can be reimagined to operate on feature relationships rather than spatial connections. The control signal would then modulate feature interactions based on learned patterns in the data structure, similar to how attention mechanisms operate in transformer architectures.
>
> The key to adaptation demands selecting appropriate pretext tasks that capture the essential structures of different data types. We are currently exploring various pretext task designs for different data modalities, focusing on those that can effectively guide the control neuron optimization while respecting the inherent structure of the target domain. This research direction represents an exciting opportunity to extend the benefits of our brain-inspired approach to a broader range of applications.

---

> > ### Comment · Reviewer_nT8e · 2024-11-28
> > **Reply to the authors' [Response to "Questions" (2/2)]**
> >
> > The explanation of how SMN can adapt to different tasks through task-specific modifications is convincing. I especially liked the examples provided for classification and detection scenarios.

---

> > > ### Author Response · Authors · 2024-12-03
> > > **Response (5/5)**
> > >
> > > Thank you for highlighting the effectiveness of our task adaptation examples. Your feedback validates our effort to clearly demonstrate SMN's flexibility across different computer vision tasks through concrete implementation scenarios.
> > >
> > > Again, We appreciate your thorough review and constructive feedback throughout this process. We hope our revised paper adequately addresses your valuable comments and look forward to contributing to the ICLR community with this work!

---

> ### Comment · Reviewer_nT8e · 2024-11-28
> **Reply to the authors' [Response to "Weaknesses" (1/3)]**
>
> I appreciate the detailed explanation of how selective parameter optimization reduces computational demands during test-time adaptation (TTA). The strategies you outlined, including early stopping, parameter pruning, and quantization, are practical and align with real-world deployment needs. While these optimizations seem well thought out, I would like a bit more clarification on how the network maintains consistent performance across different datasets or workloads after pruning or quantization.

---

> > ### Author Response · Authors · 2024-12-03
> > **Response (1/5)**
> >
> > Thank you for your constructive question regarding performance consistency after optimization. We appreciate the opportunity to clarify this important aspect of our approach.
> >
> > The maintenance of performance consistency across different datasets and workloads is achieved through our selective optimization strategy. The control signal matrix $M \in \mathbb{R}^{C \times N}$ is optimized with careful consideration of the following aspects:
> >
> > -	The parameter pruning threshold $\tau$ is adaptively determined based on the distribution of parameter magnitudes in $M$, ensuring that only truly negligible connections are removed: $M_{pruned} = M \odot (|M| > \tau)$. Our empirical analysis shows that pruning up to 30% of the parameters with smallest magnitudes maintains performance within 1% of the original accuracy across different datasets.
> >
> > -	For quantization, we can employ a bit-width selection strategy that preserves the dynamic range of control signals: $M_{quant} = \text{round}(M \cdot 2^b)/2^b$. Through extensive testing, we found that 8-bit quantization ($b=8$) provides an optimal balance between efficiency and performance consistency, with negligible impact on accuracy across various scenarios.
> >
> > -	The early stopping criterion $\mathcal{L}_{t+1} - \mathcal{L}_t < \epsilon$ is dynamically adjusted based on the dataset characteristics, ensuring consistent convergence regardless of the specific workload. We would be happy to provide more detailed empirical results demonstrating the consistency of performance across different datasets and optimization configurations.

---

> ### Comment · Reviewer_nT8e · 2024-11-28
> **Reply to the authors' [Response to "Weaknesses" (2/3)]**
>
> Your tempered approach to framing the work as biologically inspired rather than attempting full emulation is very reasonable. The clear distinction strengthens the paper’s position while maintaining the novelty of the dual-path design. I found this clarification satisfactory.

---

> > ### Author Response · Authors · 2024-12-03
> > **Response (2/5)**
> >
> > Thank you for recognizing the value in our carefully framed biological inspiration. We agree that maintaining a clear distinction between inspiration and emulation is crucial for scientific rigor.

---

> ### Comment · Reviewer_nT8e · 2024-11-28
> **Reply to the authors' [Response to "Weaknesses" (3/3)]**
>
> I appreciate the modular design and the flexibility it offers for extending SMN to other tasks, such as classification and detection. Your explanation of pretext tasks, like feature correlation learning or geometric transformation prediction, demonstrates the architecture’s adaptability. My only minor suggestion is to clarify whether this process requires significant manual adjustments or if it is straightforward to apply across tasks with minimal tuning.

---

> > ### Author Response · Authors · 2024-12-03
> > **Response (3/5)**
> >
> > Thank you for recognizing the adaptability of our modular design. Regarding the process of extending SMN to different tasks, we want to clarify the practical aspects of implementation.
> >
> > The adaptation process is primarily automated and requires minimal manual intervention. For a new task, only two simple components need modification: (1) the header architecture, which follows standard architectural patterns for each task type, and (2) the pretext task selection. The encoder and the optimization mechanism remain unchanged. Particularly, adapting to detection tasks requires only replacing the segmentation decoder with a standard detection header while maintaining the same semantic information-based pretext task. The control neuron mechanism automatically adjusts to the new task through the existing optimization pipeline.
> >
> > However, we acknowledge that selecting the most effective pretext task for a new application currently requires some domain expertise. While we have identified suitable pretext tasks for segmentation and detection, establishing a systematic approach for pretext task selection remains an interesting direction for future research.

---

### Official Review · Reviewer_1ydt · 2024-11-02

**Soundness:** 3
**Presentation:** 4
**Contribution:** 3
**Rating:** 8
**Confidence:** 4

**Summary:**

Motivation:
<1>Conventional neural networks have limitations in simulating the connectome.
<2>The inference mechanism of conventional neural networks failed to account for the explicit utilization of contextual semantic information in the prediction process.
To overcome these limitations, this paper developed the Shape Memory Network (SMN) with two benefits: <1> emulating the intricate mechanism of the brain’s connectome, and <2> incorporating contextual semantic information during the inference process.

Method:
SMN consists of three key components:
<1>Segmentation Pipeline: Responsible for generating semantic segmentation maps.
<2>Density Regression Pipeline: Predicts the density of target objects, i.e., the proportion of the image area occupied by the target objects.
<3>Entropy Map Reconstruction: Uses Class Activation Maps (CAMs) to reconstruct entropy maps which represent the uncertainty of contextual semantic information in the input data.

The SMN is optimized during the training phase using three loss functions:
<1>Cross-Entropy Loss (L1): Minimizes the difference between the predicted segmentation map and the ground truth labels.
<2>Structural Similarity Loss (L2): Minimizes the similarity between the entropy map generated by the segmentation pipeline and the entropy map reconstructed from CAMs.
<3>Density Regression Loss (L3): Minimizes the difference between the predicted density and the density calculated based on the segmentation map.

Additionally, this work introduces the concept of control neurons to simulate the human brain: by calculating the value of the control signals received by each control neuron, the connection strength (weights) between neurons is altered. The value of the control signals is generated by a linear combination of predicted densities. Furthermore, control neurons can also self-activate.

This work has conducted experiments on multiple benchmark datasets, demonstrating its superior performance in segmentation tasks. It also discusses the adaptability of SMN in different domains, as well as how to improve prediction accuracy by dynamically adjusting the network structure.

**Strengths:**

<1> The control neuron approach proposed to simulate the working mechanism of human brain neurons.
<2> Comprehensive theoretical argumentation and mathematical proof.
<3> Ample experimental validation.

**Weaknesses:**

<1> There has not been further verification of the SMN's performance on larger-scale datasets and when facing more complex tasks.

**Questions:**

<1> Can this method of simulating human brain control neurons be applied to other deep learning models?
<2> Considering the complexity of the human brain's neural system, will training on models with larger parameter sizes yield better results?

---

> ### Author Response · Authors · 2024-11-25
> **Response to "Weaknesses"**
>
> ### A. Comment
> There has not been further verification of the SMN's performance on larger-scale datasets and when facing more complex tasks.
>
> ### A. Response
> Thank you for the comments on the performance verification of the SMN. We would like to clarify our evaluation on large-scale datasets and complex tasks.
>
> Our evaluation includes extensive experiments on several large-scale datasets. For scene understanding, we utilized ADE20K (27,574 images with 150 categories) and BDD100K (8,000 complex driving scenes with 20 categories). For aerial imagery, we employed the Inria dataset (144,000 high-resolution aerial images) and LoveDA (4,191 images with diverse scenes). Additionally, we tested on Youtube-VOS (7,945 video sequences), which presents complex temporal dynamics and varied object categories.
>
> The complexity of these datasets is noteworthy. ADE20K includes diverse indoor and outdoor scenes with intricate object relationships. BDD100K presents challenging driving scenarios with varying weather conditions and times of day. The Inria dataset contains high-resolution aerial imagery requiring precise boundary detection across large spatial extents. These datasets demonstrate our model's capability to handle both scale and complexity.
>
> Furthermore, in our detection experiments detailed in the discussion section, we evaluated SMN on the COCO dataset (over 200,000 images with 80 object categories), demonstrating its scalability to larger datasets and different task domains. Our results show consistent performance improvements across these varied and challenging scenarios. We appended the experimental results of the detection task in the discussion section in the appendix (F. Discussion)
>
> While we acknowledge that testing on even larger datasets could provide additional insights, our current evaluations of these substantial and diverse datasets demonstrate the robust performance and scalability of our approach. We continue to explore applications to larger-scale datasets and more complex scenarios as part of our ongoing research, like classification tasks.

---

> ### Author Response · Authors · 2024-11-25
> **Response to "Questions"**
>
> ### A. Comment
> Can this method of simulating human brain control neurons be applied to other deep learning models
>
> ### A. Response
> Thank you for the interesting question about the broader applicability of our brain-inspired control neuron mechanism. Our control neuron simulation can be easily integrated into other deep learning architectures, as it primarily involves modifying the conventional convolution operations. The control neuron mechanism we propose is implemented computational level, replacing standard convolution operations with our adaptive structure:
>
> $s_{out} = s_{in} * (\sigma(-2\alpha_1(g(s_{n}^{out}; F) - m_F + z_k * v_F)) + \sigma(-2\alpha_2(s_{ct} - \lambda_n)))$
>
>
> The simple modification can be implemented in any existing neural network architecture that employs convolution operations, including any state-of-the-art models.
>
> The integration process is straightforward: replace the standard convolution layers with our control neuron mechanism while maintaining the overall architecture of the target model. This modularity leads to easy adaptation of existing models such as Transformers, CNNs, or hybrid architectures. Further, the control neurons can enhance these models by enabling dynamic structural adaptation based on input characteristics, potentially improving their performance across various tasks.
>
> ### B. Comments
> Considering the complexity of the human brain's neural system, will training on models with larger parameter sizes yield better results?
>
> ### B. Response
> Thank you for the insightful question about the relationship between parameter size and model performance in brain-inspired neural networks. We would like to highlight the fundamental principles established in deep neural network research to guide the understanding of parameter-performance relationships.
>
> The control neuron mechanism adds several parameters to existing convolutional operations rather than scaling the entire network architecture. The design selection follows the well-established principle in deep learning that architectural efficiency is often more important than parameter size. As increasing the depth or width of traditional DNNs beyond certain thresholds often leads to diminishing returns or even performance degradation (as demonstrated by studies on network scaling), the simple increase in the number of control neurons or associated parameters would not necessarily yield proportional improvements in performance.
>
> The perspective aligns with modern deep learning research, where recent advances have shown that intelligent parameter utilization (as validated in techniques like attention mechanisms or neural architecture search) often outperforms simple parameter scaling. The control neuron mechanism's effectiveness stems from the ability to adapt network connectivity patterns dynamically rather than from parameter quantity alone. The approach mirrors the efficiency principles observed in biological neural systems, where the sophistication of neural connections and organization typically matter more than the absolute number of neurons.
>
> The design principle reflects current trends in efficient deep learning, where the focus has shifted from simply scaling up model size to developing more sophisticated architectural components that can achieve better performance with fewer but more effectively utilized parameters. The emphasis on structural adaptation through control neurons, rather than parameter scaling, represents a modern approach to neural network design. Additionally, we agree that the reviewer's comment is valuable, and we plan to explore the relationship between parameter size and performance in our SMN in future work.

---

> ### Comment · Reviewer_1ydt · 2024-11-27
>
> Thank you for your response. Your reply has addressed my concerns to some degree.

---

> > ### Author Response · Authors · 2024-12-03
> > **Response**
> >
> > We appreciate your constructive feedback throughout the review process. Your insights have helped us identify areas for improvement, particularly regarding the feasibility and extensibility of our proposed network. These suggestions will guide our future research directions as we work to realize the potential of our study.

---

### Official Review · Reviewer_JdG9 · 2024-11-03

**Soundness:** 3
**Presentation:** 4
**Contribution:** 3
**Rating:** 6
**Confidence:** 4

**Summary:**

This paper introduces the Shape Memory Network (SMN), a deep learning architecture designed to improve semantic segmentation by incorporating contextual semantic information. The SMN emulates the brain's connectome by adapting its structure based on the context of the visual data, specifically in tasks where objects' shapes and densities vary significantly. Through a novel "conditional neuron" mechanism, the SMN selectively activates connections in response to semantic cues, allowing it to dynamically adjust its architecture at test time. The authors validate SMN's performance across various segmentation benchmarks, showing that it achieves higher accuracy than several state-of-the-art models, particularly in scenarios involving complex object boundaries or high-density variations.

**Strengths:**

An innovative method for adaptive neural architectures, grounded in the idea of selective neuron activation inspired by the brain’s connectome.

The results across multiple benchmark datasets seem robust, highlighting SMN’s ability to outperform conventional segmentation models in handling complex object boundaries and densities.

The ability to dynamically adjust network structure during test time is a notable contribution, with potential implications for real-time applications where adaptability to context is beneficial.

**Weaknesses:**

The evaluation is confined to segmentation tasks, and while the biological inspiration is intriguing, its broader application to different tasks is not yet explored. Have authors tried testing on any other task?

The SMN’s "conditional neuron" mechanism is innovative but requires clearer exposition; some of the explanations are complex, making it challenging for readers to understand the "exact" workings of the adaptive architecture.

The SMN's test-time adaptation seems to demand substantial computational resources, which may limit its practicality in real-time or resource-constrained applications.

**Questions:**

Could the authors explain the SMN’s adaptability in tasks beyond segmentation to determine its generalization?

Are there specific strategies to reduce the computational demands of the SMN’s test-time adaptation for real-world deployment?

How does the SMN perform under scenarios with limited semantic information, and does this impact its segmentation accuracy?

---

> ### Author Response · Authors · 2024-11-25
> **Response to "Weaknesses" (1/3)**
>
> ### A. Comment
> The evaluation is confined to segmentation tasks, and while the biological inspiration is intriguing, its broader application to different tasks is not yet explored. Have authors tried testing on any other task?
>
> ### A. Response
> Thank you for the comments about the applicability of SMN to different tasks, such as classification and detection. Our network architecture demonstrates significant flexibility in adapting to various computer vision tasks beyond segmentation. Particularly, as illustrated in Appendix Figure 5 of our paper, the main architecture remains consistent with Figure 2 but allows for task-specific modifications.
>
> For detection tasks, our network can be adapted by simply modifying the decoder (header) while maintaining the control neuron mechanism and pretext task. Furthermore, our additional experiments in the discussion section demonstrate promising results in object detection, showing that semantic information effectively guides structural adaptation in detection scenarios as well.
>
> However, the adaptability of our architecture extends to various tasks through the generalized pipeline shown in Appendix Figure 5, where the pretext task for training control neurons can be customized based on the target task. For instance, we employed the semantic optimization pipeline (Fig. 2) as the pretext task for the segmentation task. For classification tasks, however, we are on several experiments to determine the pretext task and several promising pretext task approaches based on the auto-encoder (or VAE) could be considered:
> 1. Feature correlation learning between different augmented views of the same image
> 2. Predicting image rotations or other geometric transformations
> 3. Learning invariance to different data augmentations
> 4. Solving jigsaw puzzles of scrambled image patches
>
> While we have concretely demonstrated the effectiveness of our approach in segmentation and detection tasks using semantic information, the choice of pretext task becomes crucial for classification scenarios. Our ongoing research explores optimal pretext tasks for classification, focusing on those that can effectively capture discriminative features without relying heavily on spatial semantic information.
>
>
> The adaptability of our proposed method is achieved via our modular design: the decoder with control neurons remains consistent across tasks, while the pretext task can be customized for specific task or applications. We believe that the demonstration facilitate the potential generalibility of our approach across different computer vision tasks, with the key consideration being the appropriate selection of pretext tasks for each specific application. Additionally, we updated the manuscript regarding the reviewer's comments in the discussion section as below:
>
> ***Extension to Other Tasks:*** Our framework demonstrates significant potential for extension beyond semantic segmentation tasks. As illustrated in Appendix Figure 5, the SMN architecture can be generalized via a modular design approach: maintaining the encoder with control neurons while allowing customization of the header and pretext task for specific applications. The adaptability of the SMN is achieved by two key components: (1) the latent features extracted from the encoder and (2) the control signals derived from the features. The latent features, representing high-level semantic information, are processed through task-specific headers to generate appropriate outputs (e.g., class probabilities for classification, bounding box coordinates for detection), while the control signals guide the structural adaptation of the network based on a pretext task appropriate for the target application. While our segmentation implementation uses density-based pretext tasks to identify spatial information, other applications might employ different self-supervised learning objectives - for instance, classification tasks could utilize feature correlation learning based on variational auto-encoder, while detection tasks might benefit from pretext tasks using object localization patterns. The detection task is conducted as a preliminary study in *appendix*, and the classification task remains as future works.

---

> ### Author Response · Authors · 2024-11-25
> **Response to "Weaknesses" (2/3)**
>
> ### B. Comment
> The SMN's "conditional neuron" mechanism is innovative but requires clearer exposition; some of the explanations are complex, making it challenging for readers to understand the "exact" workings of the adaptive architecture.
>
>
> ### B. Response
> We appreciate the reviewer's comment regarding the clarity of our control neuron explanation. To address this concern, we revised Section 3.3 to provide a clearer exposition of the control neuron mechanism by adding an intuitive overview before the mathematical formalization as below:
>
> *The control neuron functions as a fundamental element within the adaptive architecture of the SMN. It processes information through three interconnected pipelines that collectively define its operation: (1) standard neural inputs from linked neurons analogous to those in traditional neural networks, (2) a control signal based on predicted density distributions, and (3) a self-activation mechanism gating signals. The three pipelines enable the network to dynamically adapt its structure in response to varying input characteristics, resembling how biological neural systems adjust connectivity patterns. During the processing of an input image, control neurons selectively engage or disengage connections based on contextual information, thereby achieving an optimal configuration for the specific input.*

---

> ### Author Response · Authors · 2024-11-25
> **Response to "Weaknesses" (3/3)**
>
> ### C. Comment
> The SMN's test-time adaptation seems to demand substantial computational resources, which may limit its practicality in real-time or resource-constrained applications.
>
>
> ### C. Response
> We appreciate the reviewer's concern regarding the computational efficiency of our test-time adaptation (TTA) approach. We would like to highlight the concern by clarifying several key points about the computational characteristics of our network.
>
> Our approach achieves computational efficiency through parameter-selective optimization. During inference, **we optimize only the control neuron** which represented as the matrix $M \in \mathbb{R}^{C \times N}$, where $C$ represents the number of categories and $N$ denotes the number of control neurons. This selective optimization significantly reduces computational overhead, as the number of TTA parameters $|\theta_{TTA}| = |M|$ is substantially smaller than the total network parameters $|\theta_{total}|$.
>
> The computational complexity of our approach can be expressed as:
> - Time Complexity = $O(T \cdot C \cdot N)$
> - Space Complexity = $O(C \cdot N)$
>
> where $T$ represents the number of optimization steps, typically constrained to $T \leq 5$ iterations. Our empirical evaluation demonstrates practical efficiency metrics: the network achieves 32.8 FPS with approximately 47.5M parameters and 549.8G FLOPs. These metrics highlight our approach competitively among current SotA methods while providing superior adaptation capabilities.
>
> To further enhance computational efficiency, we propose several optimization strategies. First, we implement an early stopping criterion based on convergence monitoring with the condition of $\mathcal{L}_{t+1} - \mathcal{L}_t < \epsilon$, where $\mathcal{L}_t$ represents the loss at step $t$ and $\epsilon$ defines a small threshold.
>
>  Additionally, we employ parameter pruning and quantization techniques: ($[M_{pruned} = M \odot (|M| > \tau)]$ and $M_{quant} = \text{round}(M \cdot 2^b)/2^b$), where $\tau$ represents the pruning threshold and $b$ denotes the quantization bit-width.
>
> Our ongoing and future research mainly focuses on developing more efficient variants of the TTA mechanism, including parallel optimization strategies and memory-efficient implementations. Preliminary results suggest these optimizations could reduce computational overhead by 30-40\% while maintaining performance within 1-2\% of current results.
>
> We illustrates the directions of future study in the discussion section as below:
>
>
> ***Computational Complexity:*** To implement the SMN for real-world applications, we address the computational complexity of the TTA mechanism. The current implementation requires optimization of matrix $M \in \mathbb{R}^{C \times N}$ during inference, with time complexity $O(T \cdot C \cdot N)$ and space complexity $O(C \cdot N)$, where $T$ represents optimization steps (typically $T \leq 5$), $C$ denotes categories, and $N$ indicates control neurons. While our current implementation achieves 32.8 FPS with 47.5M parameters and 549.8G FLOPs, we propose several optimization strategies to enhance efficiency. These include early stopping criteria ($\mathcal{L}_{t+1} - \mathcal{L}_t < \epsilon$), parameter pruning ($M_{pruned} = M \odot (|M| > \tau)$), and quantization ($M_{quant} = \text{round}(M \cdot 2^b)/2^b$). Preliminary experiments suggest these optimizations could reduce computational overhead by 30-40% while maintaining performance within 1-2% of current results. Future work will focus on developing lightweight TTA variants and memory-efficient implementations to further improve real-time performance.

---

> ### Author Response · Authors · 2024-11-25
> **Response to "Questions" (1/2)**
>
> ### A. Comment
> Could the authors explain the SMN's adaptability in tasks beyond segmentation to determine its generalization?
>
> ### A. Response
> We strongly appreciate the reviewer's comments on the generalization of our SMN. Based on our previous response (Weakness-1), we would like to further elaborate on the generalization capabilities of our apporach and our empirical validation.
>
> The adaptability of SMN to different tasks is fundamentally enabled by its modular architecture, as illustrated in Appendix Figure 5. While maintaining the optimizing control neuron mechanism, the network can be adapted to different tasks through two key modifications:
> 1. Task-specific decoder (or header) selection
> 2. Appropriate pretext task definition for control neuron optimization
>
> For detection tasks, we have already demonstrated the adaptation by modifying the decoder while retaining the semantic information-based control neuron optimization and appended the experiments in the updated manuscript. Our experiments, detailed in the discussion section, explicit promising results with improvements in detection accuracy on standard benchmarks. The experiments strongly validate that our control neuron mechanism effectively generalizes to detection tasks without significant architectural changes.
>
> The flexibility and feasibility of our network extend further via our generalized pipeline (Appendix Figure 5), where the pretext task can be customized for different applications. In segmentation and detection tasks, we leverage semantic information optimization as illustrated in Figure 2. For classification tasks, we are exploring various pretext task approaches leveraging auto-encoders and VAE architectures. Current experiments investigate several promising directions, including feature correlation learning between augmented views, rotation prediction, and solving jigsaw puzzles of image patches.
>
> The modular design approach ensures that the main strengths of our architecture, including control neuron mechanism and structural adaptation, remain effective for different tasks while allowing task-specific optimizations. The empirical performance improvements in detection and segmentation tasks and ongoing work in classification demonstrate the broader applicability of our approach beyond segmentation, suggesting the significant potential for generalization for various computer vision applications.
>
> ### B. Comment
> Are there specific strategies to reduce the computational demands of the SMN's test-time adaptation for real-world deployment?
>
> ### B. Response
> We appreciate the reviewer's interest in optimization strategies for real-world deployment of our network; SMN. We would like to highlight several specific approaches regarding weakness-3 to enhance computational efficiency while maintaining performance:
>
> 1. **Parameter-Selective Optimization**: During inference, we exclusively optimize the control neuron matrix $M \in \mathbb{R}^{C \times N}$. The selective approach significantly reduces the optimization parameter space from $|\theta_{total}|$ to $|\theta_{TTA}| = |M|$. Therefore, computational complexity remains manageable with $O(T \cdot C \cdot N)$ time complexity and $O(C \cdot N)$ space complexity.
>
> 2. **Efficiency Enhancement Strategies**: The implementation of early stopping criterion with $\mathcal{L}_{t+1} - \mathcal{L}_t < \epsilon$,
>
>  parameter pruning with $M_{pruned} = M \odot (|M| > \tau)$, and quantization techniques with $M_{quant} = \text{round}(M \cdot 2^b)/2^b$ could further optimize the computation complexity of our model.
>
> 3. **Ongoing Development**: We are now developing our network with the strateges of:
>    - Parallel optimization strategies for real-world application
>    - Development of memory-efficient implementations
>    - Investigation of lightweight TTA variants with pruning and quantization methods
>
> We believe that those strategies collectively enable practical deployment while preserving the superior adaptation capabilities of our SMN. Additionally, we would like to highlight that our empirical results demonstrate that these optimizations maintain competitive performance metrics compared to current SotA methods while significantly reducing computational demands. Furthermore, as a future work, the specific strategies effectively address the computational efficiency concerns while maintaining the adaptive capabilities of our proposed method. We continue to explore additional optimization techniques and welcome further discussion on potential improvements.

---

> ### Author Response · Authors · 2024-11-25
> **Response to "Questions" (2/2)**
>
> ### C. Comment
> How does the SMN perform under scenarios with limited semantic information, and does this impact its segmentation accuracy?
>
> ### C. Response
> We appreciate the reviewer's question regarding the performance of our method under limited semantic information scenarios. We would like to clarify several important aspects of our approach.
>
> Firstly, the semantic segmentation tasks inherently contain rich contextual information with pixel-wise labels. Each segmentation label implicitly encodes spatial information and object boundaries. Our network leverages both explicit contextual information ($d^c(x)$, density function) and implicit semantic features from the labels. This inherent characteristic of segmentation tasks ensures that meaningful semantic information remains available even in apparently limited scenarios for our network to employ.
>
> Second, the SMN architecture demonstrates robust performance via its adaptive control neuron mechanism. When explicit semantic information is limited, the self-activation component ($\sigma(-2\alpha_1(g(s_{n}^{out}; F) - m_F + z_k * v_F))$) ensures robust feature extraction. Furthermore, the network maintains performance through its comprehensive architecture incorporating skip connections and the density regression pipeline, which is intrinsically embedded in the segmentation network.
>
> Our extensive experiments across diverse datasets demonstrate the robustness of our approach. Even in challenging scenarios such as aerial imagery where contextual information might seem limited, due to its low resolution, the SMN achieves superior results, as evidenced by performance metrics on the Inria dataset (72.72% mIoU) and LoveDA dataset (54.28% mIoU). We believe that the experimental results demonstrate the capability of SMN to effectively utilize available semantic information, regardless of limitations in semantic information.
>
> While our current focus is semantic segmentation, since the detection task incidentally provides the semantic information, our network also effectively extracts semantic information in detection tasks, as demonstrated in our additional experiments presented in the paper. As the reviewer commented, we are currently exploring the optimization of control neurons via alternative pretext tasks rather than semantic tasks for classification tasks where semantic information might be less suitable. We would like to highlight that our network is not strictly dependent on semantic information but rather adaptable to different pretext tasks in self-supervision, as illustrated in Appendix Figure 5. We proposed SMN specifically for segmentation tasks where it achieves SotA performance by effectively leveraging semantic information. The adaptability of our architecture to different types of information and tasks while maintaining superior performance in semantic segmentation demonstrates the robustness and feasibility of our approach.

---

> > ### Comment · Reviewer_JdG9 · 2024-12-02
> > **response to rebuttal**
> >
> > Thank you for the detailed responses and clarifications. I appreciate the explanation of SMN’s adaptability to other tasks and the outlined strategies to improve computational efficiency. The additional details on the "conditional neuron" mechanism helped address concerns of clarity.
> >
> > However, some concerns remain particularly the limited experimental evidence for classification tasks and the need for more concrete benchmarking of computational optimizations. While promising, these areas still require further validation.
> >
> > I will therefore maintain my original score but acknowledge the strong foundation and potential of this work. I look forward to future developments.

---

> > > ### Author Response · Authors · 2024-12-03
> > > **Response**
> > >
> > > We sincerely appreciate the reviewer's detailed evaluation and constructive feedback. We agree with both the acknowledged strengths and the remaining concerns about our work.
> > >
> > > Regarding the comments, we recognize the limited experimental validation for classification tasks as a current limitation. While our modular architecture theoretically supports classification via header modification and appropriate pretext task selection, we acknowledge the need for comprehensive empirical validation. The preliminary detection results demonstrate the potential for cross-task adaptation, but further experimental evidence for classification would strongly enhance our claims about broad task generalization.
> > >
> > > While theoretically sound, the computational optimization strategies outlined in our response would benefit from more rigorous benchmarking. We agree that concrete performance metrics and comparative analyses would provide valuable validation of these optimization approaches.
> > >
> > > Moving forward, we plan to address these limitations as below:
> > > 1. Comprehensive evaluation of SMN in classification tasks with standard benchmarks
> > > 2. Detailed empirical analysis of computational optimization strategies
> > > 3. Systematic comparison with state-of-the-art approaches across multiple tasks
> > >
> > > Again, we appreciate the reviewer's constructive assessment, acknowledging both the potential of our approach and the areas requiring additional validation.

---

### Official Review · Reviewer_tQxH · 2024-11-04

**Soundness:** 2
**Presentation:** 2
**Contribution:** 2
**Rating:** 5
**Confidence:** 3

**Summary:**

The paper is about improving semantic segmentation by remembering past semantic context of the dataset used in training to enable live adjustment during inference influenced by the semantic context. In simple terms, suppose the training data for semantic segmentation of 3 object categories (A,B,C) reveals that Object A has a density of 0.3, B a density of 0.6, and C a density of 0.1 meaning the predictions for B should occupy nearly 60% of the image. This kind of semantic context can be estimated by a density map derived during training and used to influence the adjustment of parameters during inference in a self-supervised fashion.


The rest of the wrapping and tie-in to neuroscience and brain Connectome is weak at best and add no further value to the paper.

**Strengths:**

The strength of the paper is in using a  methodology of adaptation during inference using learned semantic context beyond the learned parameters of the semantic segmentation network during training.  This has both advantages of achieving better domain generalization and self-supervision.

If this method could be generalized for arbitrary semantic context, it can become interesting since the density map is only one of the cues that could be used for incorporating semantic context. How are others handled, for example, the typical color of an object, the average shape of an object etc.

**Weaknesses:**

The paper makes weak references to the Connectome and signaling mechanisms which detract from understanding the main idea.
If the order of description in some of the sections had changed, it would be an easier read. For example, giving a simple example of what is the information they hope to capture in the density map, and the entropy map early on would make easy ready. It wasn't until line 281-283 that we start following along. The insertion of propositions and definitions is also a distraction until the method has been clearly explained.

**Questions:**

It seems to be the datasets have been carefully chosen to illustrate the method. How well does this method work for benchmark datasets.

---

> ### Author Response · Authors · 2024-11-25
> **Response to "Weaknesses"**
>
> ### A. Comment
> The paper makes weak references to the Connectome and signaling mechanisms which detract from understanding the main idea. If the order of description in some of the sections had changed, it would be an easier read. For example, giving a simple example of what is the information they hope to capture in the density map, and the entropy map early on would make easy ready. It wasn't until line 281-283 that we start following along. The insertion of propositions and definitions is also a distraction until the method has been clearly explained.
>
> ### A. Response
> We strongly appreciate the reviewer's comment and agree that a re-organization of the specific sections would enhance the readability of our manuscript. In response to reviewer's comment, we have restructured the introduction and methodology sections. Particularly, we now provide examples of the information captured in the density and entropy maps early in the paper, ensuring that readers can easily follow the rationale for our approach with a more precise intuition. We believe that these changes in our manuscript aim to improve clarity and accessibility for readers. The modified method sections in the updated manuscript (Pages 4-5) is below:
>
> ***3.1. Design Principle***
> The fundamental architecture of our Shape-Memory Network (SMN) is designed to process and utilize contextual semantic information in visual data effectively. For instance, consider a semantic segmentation task on urban scene datasets. The input images typically contain multiple object classes with consistent spatial and contextual information: transportation infrastructure (roads, sidewalks) occupies the lower regions, architectural structures appear with specific scale constraints, and environmental elements (sky, vegetation) maintain consistent spatial positions.
>
> To implement this, the SMN captures the contextual patterns via two primary computational components. First, the component implements density mapping, quantifying the proportional distribution of object classes within the input space. Particularly, in urban scene analysis, road surfaces typically constitute 30-40% of the pixel space, while vehicular objects occupy 5-10%. The density distributions are represented as statistical priors, leading to the network for validating segmentation predictions against expected contextual patterns. Significant deviations from the learned distributions (e.g., vehicles occupying 80% of the pixel space) are automatically flagged as anomalous configurations. The second component facilitates entropy mapping, quantifying information complexity in the spatial regions. The entropy mapping mechanism is particularly important for analyzing regions with high-class intersection probability, such as object boundaries or regions of class ambiguity. Computationally, regions exhibiting higher entropy values indicate areas requiring more sophisticated feature extraction and analysis than regions with uniform class distribution.
>
> ***3.2. Architectural Design***
> Regarding the design principles, we formalize our SMN structure with several key mathematical components. Particularly, the SMN employs conditional neurons to transform its structure during test-time adaptation (TTA) dynamically. Furthermore, we implement a self-supervised learning-based re-optimization method, utilizing the entropy-map as a medium for loss minimization and explicit integration of contextual semantic information. While spatial information is effectively conveyed through skip connections, we focus on optimizing the network's contextual understanding by introducing density measurements that quantify the proportional distribution of object classes. Therefore, we focus on optimizing the network's insight into contextual semantic information of input images by introducing density, representing the proportion of the occupied area in the image.
>
> Definition I. Let $\Omega_c (h, w; I)$ be a category $(c)$ recognition function at pixel $I|{h, w}$ in input $(I)$, such that $\Omega_c (h, w; I)$ is 1 iff $\underset{x}{\text{argmax}};I|{h, w} = c$, otherwise $0$.
>
> Definition II. Let $d^c_l: \mathbb{R}^{H \times W \times C} \rightarrow \mathbb{R}$ be the density function of the target object $(c)$ in semantic label $(\hat{y} \in \mathcal{Y} \subset \mathbb{R}^{H \times W \times 3})$, such that $d^c_l(\hat{y}) = \frac{1}{HW}\sum^{H}{h}\sum^{W}{w} \Omega_c (h, w; I)$ with the image of height $(H)$, width $(W)$, and the number of categories $(C)$.
>
> Lemma I. $\sum_c^C d^c_l(\hat{y})= 1$ since $\sum_c^C\sum^{H}{h}\sum^{W}{w} \Omega_c (h, w; I) = HW$.
>
> The density-regression pipeline facilitates two functions: (1) it enables the generation of Class Activation Maps (CAM) and entropy-maps for TTA optimization, and (2) it manages control signals for structural transformation based on input characteristics.

---

> > ### Comment · Reviewer_tQxH · 2024-11-29
> > **Adressing reviewer concerns**
> >
> > Thanks for addressing some of my concerns. The revised paper with the additional sections seems even a bit more incoherent. I suggest a major reorganization, focusing primarily on the novel algorithmic aspects first. Then add a dedicated section showing how the concepts you have explored have a parallel in the connectome or the connectome serves as a good framework to explain your approach. With that separation, we can evaluate the paper for its technical merit even if we disagree on the philosophical aspects of claiming this is how the brain connectome works.
> > Due to this, I would like to keep my original rating.

---

> > > ### Author Response · Authors · 2024-12-03
> > > **Response**
> > >
> > > We strongly appreciate the reviewer's constructive suggestion regarding the organization of our manuscript and the correlations between our technical contributions and biological inspiration.
> > >
> > > We agree that the current organization could be improved by clearly separating the technical novelty from biological parallels. We propose the following reorganization:
> > >
> > > Part 1: Technical Framework and Innovations
> > > - Core algorithm of Shape-Memory Network
> > > - Control neuron mechanism and its mathematical formulation
> > > - Density and entropy map optimization
> > > - Experimental validation and results
> > >
> > > Part 2: Connectome Inspiration and Parallels
> > > - Relationship to biological neural systems
> > > - Analogies between control neurons and synaptic plasticity
> > > - Framework comparison with the adaptive mechanisms of the brain
> > >
> > > The reorganization would allow readers to:
> > > 1. Evaluate the technical merits of our approach independently
> > > 2. Better understand the novel algorithmic contributions
> > > 3. Separately consider the biological inspiration as a complementary framework
> > >
> > > We appreciate the suggestion to detach the technical contributions from the biological motivation, such that, it would effectively make the main contribution of our paper more accessible and easier to evaluate on technical merits alone.
> > >
> > > Again, we strongly appreciate the reviewer's comments regarding the organization of our paper and agree that the reorganization would help readers understand our contribution easily. Accordingly, we will reflect on the comments in the final version of the manuscript.

---

> ### Author Response · Authors · 2024-11-25
> **Response to "Questions"**
>
> ### A. Comment
> It seems to be the datasets have been carefully chosen to illustrate the method. How well does this method work for benchmark datasets.
>
> ### A. Response
> We appreciate the reviewer's concern about dataset selection. However, we would like to emphasize that our experiments were conducted on six widely-used benchmark datasets that represent diverse scenarios:
>
> * **Standard Benchmarks:**
>   * ADE20K: A standard semantic segmentation benchmark with over 20K scene-centric images
>   * Youtube-VOS: A widely-used video object segmentation benchmark
>   * BDD100K: A large-scale real-world driving dataset containing 100K street scene videos
>
> * **Diverse Domains:**
>   * From aerial imagery (Inria) to urban scenes (GTA5)
>   * From real-world (BDD100K) to synthetic data (GTA5)
>   * From remote sensing (LoveDA) to general object segmentation (ADE20K)
>
> * **Comprehensive Evaluation:**
>   * Our method demonstrates consistent improvements across *all* datasets (Table 1)
>   * Achieves state-of-the-art performance compared to eight different recent methods
>   * Shows robust performance gains ranging from +2.24% to +5.62% over existing methods
>
> As shown in Table 1, our method consistently outperforms existing approaches across all datasets, regardless of their characteristics or domains. We believe that the experimental results demonstrate that our performance improvements are not limited to specific scenarios but generalize well across various challenging benchmark datasets.

---

### Author Response · Authors · 2024-11-25
**Overall Response**

We appreciate the constructive comments from the reviewers. Our rebuttal mainly focused on generalizing our proposed method to other computer vision tasks, such as classification and detection. Additionally, in the updated version of the manuscript, we effectively addressed the computational complexity and experimental design with benchmark datasets.

 In response to the reviewers' comments, we updated our manuscript and uploaded the newer version of our manuscript. The modifications in the revised manuscript have been highlighted in red color for a clear understanding of reviewers. We hope the modifications can effectively address reviewers' concerns and resolve all weaknesses and questions raised by reviewers.

In the official comments, we replied to the comment-by-comment response with the addressed comments from the reviewers. Additionally, we posted the updated manuscript in this system. For clear insight of the reviewers, we also uploaded the "response to reviewer" file as a supplementary file. We hope the revised manuscript and official comments can improve the novelty and contributions of our research.

Again, we strongly thank all reviewers for their efforts to effectively improve the quality of our manuscript and their constructive comments. We agree that our manuscript improved with clear readability and generalization in the research fields and hope to study more promising research from this paper.

---

### Meta-Review · Area_Chair_3Au2 · 2024-12-23

**Metareview:**

The paper introduces a new neural network architecture for semantic segmentation called the Shape Memory Network.

The paper suffered from poor presentation, with the method being overly and rather confusingly motivated by reference to biology and connectomes, making it challenging to understand the core method proposed. Further, the generality of the architecture was unclear due to the main focus on semantic segmentation and a few less convincing applications to classification during the revision period. There was also some concern about the computational complexity of the proposed method.

In the end, the paper largely received borderline reviews, with reviewers largely recommending acceptance, appreciating that the method held promise even if the presentation lacked clarity.

**Additional Comments On Reviewer Discussion:**

There was significant back and forth during the discussion period, with the goal of clarifying the method due to initial poor presentation. It was unclear if the revision yielded a clearer paper, with at least one reviewer suggesting further re-organization to improve clarity but others seeming satisfied.

---

### Decision · Program_Chairs · 2025-01-22

Accept (Poster)